

# 1  A systematic review of climate change science relevant to

# 2  Australian design flood estimation

Conrad Wasko[1], Seth Westra[2], Rory Nathan[1], Acacia Pepler[3,4], Timothy H. Raupach[4,5,6], Andrew
Dowdy[3,4,5], Fiona Johnson[7,8], Michelle Ho[1], Kathleen L. McInnes[9], Doerte Jakob[10], Jason
Evans[4,5,6], Gabriele Villarini[11,12], Hayley J. Fowler[13]
[1]Department of Infrastructure Engineering, The University of Melbourne, Parkville, Victoria, Australia
[2]School of Architectural and Civil Engineering, University of Adelaide, Adelaide, Australia
[3]Australian Bureau of Meteorology, Sydney, Australia
[4]National Environmental Science Program Climate System Hub, Australia
[5]Climate Change Research Centre, University of New South Wales, Sydney, New South Wales, Australia
[6]ARC Centre of Excellence for Climate Extremes, University of New South Wales, Kensington, New South Wales,
Australia
[7]Water Research Centre, School of Civil and Environmental Engineering, University of New South Wales,
Kensington, New South Wales, Australia
[8]Australia Research Council Training Centre in Data Analytics for Resources and Environments
[9]CSIRO Environment, Aspendale, Australia
[10]Australian Bureau of Meteorology, Melbourne, Australia
[11]Department of Civil and Environmental Engineering, Princeton University, New Jersey, USA
[12]High Meadows Environmental Institute, Princeton University, New Jersey, USA
[13]School of Engineering, Newcastle University, Newcastle upon Tyne, UK
*Correspondence to*: Conrad Wasko (conrad.wasko@unimelb.edu.au)
**Abstract**
In response to flood risk, design flood estimation is a cornerstone of planning, infrastructure design, setting of
insurance premiums and emergency response planning. Under stationary assumptions, flood guidance and the methods
used in design flood estimation are firmly established in practice and mature in their theoretical foundations, but under
climate change, guidance is still in its infancy. Human-caused climate change is influencing factors that contribute to
flood risk such as rainfall extremes and soil moisture, and that there is a need for updated flood guidance. However, a
barrier to updating flood guidance is the translation of the science into practical application. For example, most science
focuses on examining trends in annual maximum flood events, or the application of non-stationary flood frequency
analysis. Although this science is valuable, in practice design flood estimation focuses on exceedance probabilities
much rarer than annual maximum events, such as the 1% annual exceedance probability event or even rarer, using
rainfall-based procedures, at locations where there are little to no observations of streamflow. Here, we perform a



systematic review to summarise the state-of-the-art understanding of the impact of climate change on design flood
estimation in the Australian context, while also drawing on international literature. In addition, a meta-analysis,
whereby results from multiple studies are combined, is conducted for extreme rainfall to provide quantitative estimates
of possible future changes. This information is described in the context of contemporary design flood estimation
practice, to facilitate the inclusion of climate science into design flood estimation practice.

### 1. Introduction

Flood assessment provides critical information to evaluate the tolerability or acceptability of flood risks, and to support
the development of risk management strategies. Flood risk reduction measures can be exercised through the
construction of flood mitigation structures, zoning and development controls, and non-structural measures to better
respond to floods when they do occur, for example through flood warning systems and emergency management
planning. For hereon we adopt the term 'risk' to mean flood risk. Across the world, the associated hypothetical flood
adopted for design and planning purposes for management of risk is termed the *design flood* (Jain and Singh, 2003).
In Australia, the design flood is characterised in terms of an annual exceedance probability (AEP) rather than an annual
recurrence interval (ARI) with the aim of better highlighting the annual risks that the community is exposed to. There
are many different methods of estimating the design flood applicable for different AEPs, ranging from *flood frequency*
*analysis* which use streamflow observations, to *continuous simulation* which use long sequences of rainfall
observations, to those that use rainfall in *event-based modelling* through Intensity-Duration-Frequency (IDF) curves
(in Australia termed Intensity-Frequency-Duration, or IFD curves) and/or Probable Maximum Precipitation (PMP) as
inputs. Methods of design flood estimation are commonly stipulated by guiding documents; for example, The
Guidelines of Determining Flood Flow Frequency – Bulletin 17C (England et al., 2019) in the U.S.A., the Flood
Estimation Handbook (Institute of Hydrology, 1999) in the UK, and Australian Rainfall and Runoff (Ball et al., 2019a)
in Australia. Such guidance documents, though not necessarily legally binding, are seen as representing best practice.
Traditionally, the AEP, or flood quantile to which it corresponds, has been assumed to be static; however, with climate
change, it is now recognised that the flood hazard is changing (Milly et al., 2008). A recent review of climate change
guidance has found that several jurisdictions around the world are already incorporating climate change into their
design flood guidance (Wasko et al., 2021b). For example, Belgium, Denmark, England, New Zealand, Scotland,
Sweden, the UK, and Wales are all recommending the use of climate change adjustment factors for IFD rainfall
intensities. Many countries also recommend higher climate change adjustment factors for rarer precipitation events,
consistent with findings from various modelling studies that rarer events will intensify more with climate change
(Gründemann et al., 2022; Pendergrass and Hartmann, 2014). Shorter duration storms are likely to intensify at a greater
rate than longer duration storms (Fowler et al., 2021) and subsequently, some guidance, such as that from New Zealand
and the UK, also accounts for storm duration in their climate change adjustment factors (Wasko et al., 2021b).
Although substantial advances have been made in adjusting design flood estimation guidance to include climate
change, there remains a disconnect between climate science and existing guidance. For example, although there are
climate change adjustment techniques available for generating altered precipitation inputs, none of the guidance
reviewed provided recommendations for adjusting rainfall sequences used in continuous simulation. Also, current



guidelines for estimation of probable maximum precipitation (PMP) assume a stationary climate (Salas et al., 2020) despite evidence to the contrary (Kunkel et al., 2013; Visser et al., 2022). Finally, while research has been undertaken into non-stationary flood frequency analysis, and the methods are relatively mature (Salas et al., 2018; Stedinger and Griffis, 2011), these have not been adopted in guidance. For example, Bulletin 17C assumes time-invariance (England et al., 2019).

There are multiple reasons for the disconnect between the science and flood estimation practice. Although widely accepted in the scientific literature, the "chain-of-models" approach – whereby General Circulation Model (GCM) outputs are bias corrected and downscaled to create inputs for hazard modelling (Hakala et al., 2019) – has large uncertainties (Kundzewicz and Stakhiv, 2010; Lee et al., 2020), with the uncertainties often seen as a barrier for adoption (Wasko et al., 2021b). There are also disconnects between the methods employed in flood estimation and the climate science, with little research undertaken on the non-stationarity of other factors affecting the design flood estimate other than the peak rainfall depth (i.e. IFDs), such as the temporal and spatial pattern of rainfall or the influence of antecedent conditions on rainfall losses (Quintero et al., 2022). Finally, most climate science focuses on the annual maximum daily precipitation, often referred to as the 'RX1 day index' or Rx1D (Zhang et al., 2011), to measure changes in extremes, with standard climate models not adequately resolving the processes that govern sub-daily rainfall extremes. In contrast, design flood estimation generally requires consideration of sub-daily rainfall totals and events much rarer than annual maxima.

With a literature search finding no existing synthesis of climate science relevant to the specific needs of design flood estimation, here we undertake a systematic review of the latest science directly relevant to the inputs used in design flood estimation. Although we focus on science relevant to Australia, international literature is incorporated, as design flood estimation methods are used around the world. Finally, we combine the results from individual studies using the process of meta-analysis to assess the level of consensus of different sources of evidence relating specifically to the design flood estimation input of extreme rainfall under climate change. This review represents a critical step in updating flood guidance and translating scientific knowledge into design flood practice. This review aims to (a) serve as a template for scientific reviews as they relate to design flood estimation guidance updates, and (b) identify knowledge gaps in the scientific literature that are required by engineers who perform design flood estimation.

## 2. Background to design flood estimation practice

Common to all design flood estimation methods is the conversion of empirical data (either at-site or from analogous regions) to probability estimates, with the primary differences between methods relating to where in the causal chain of flooding the data are obtained, and where the probability model is fitted. To contextualise the systematic review this section briefly introduces the primary design flood estimation approaches, with Figure 1 showing the typical AEP that each method applies to.

**1. Flood frequency analysis (FFA)**: A flood frequency curve is derived by fitting a probability distribution such as an extreme value distribution to streamflow data, which is then subsequently used to estimate the design flood quantiles. This method is limited to catchments where streamflow data is available unless data can be transposed or





corrected. As flood records are typically in the order of decades, AEPs rarer than 1 in 50 are generally subject to
considerable uncertainty. Hence, flood frequency analysis is often not used by practitioners as either at-site data is
unavailable, the record is too short to estimate the target quantile, or there have been significant changes to the
catchment over the period of record. Regional flood frequency analysis is an extension of flood frequency analysis
where space is traded for time by pooling regional data to extend the applicability of this method to rarer events.
**2. Continuous simulation:** Where long rainfall records are available, it may be possible to use a hydrologic model to
simulate the streamflow of a catchment, at which point flood maxima are then extracted from the modelled output to
derive flood quantiles using an appropriate probability model. Where long rainfall records are not available, the
modelling can be forced by stochastically generated data. This approach is very useful in joint probability assessments
where system performance varies over multiple temporal and spatial scales (e.g., multiple sewer overflows or the
design of linear infrastructure), or in more volume-dependent systems comprised of compound storages. Due to its
reliance on long rainfall sequences, continuous simulation, like flood frequency analysis, is only usually used to
estimate more frequent flood events.
**3. Event-based (IFD) modelling:** This is the most common method used for design flood estimation. A rainfall depth
or intensity of given AEP and duration is sampled from an IFD curve and combined with the rainfall temporal patterns
to create a design rainfall event (or "burst") of a given duration. In some applications, it is preferable to consider
design events based on complete storms, and thus it is necessary to augment the rainfall bursts derived from IFD
curves with rainfalls that might be expected to occur prior (or subsequent) to the burst period. As the design storm
rainfall is generally a point rainfall but applied over a catchment, an Areal Reduction Factor (ARF) is applied before
the design event is used as an input to a model to estimate the runoff hydrograph. Rainfall that does not contribute to
the flood hydrograph as it enters depressions in the catchment, is intercepted, or is infiltrated into the soil, is removed
through a "loss" model. Finally, the hydrograph response may be modulated by the tail water conditions, where the
sea level will modulate the catchment outflow.
Due to the severe consequences of failures, critical infrastructure, such as dams or nuclear facilities, often need to be
designed to withstand the largest event that is physically plausible, termed the Probable Maximum Flood (PMF). Like
the above event-based modelling description, the PMF is derived from a rainfall event, but in this case the rainfall is
the PMP. Most local jurisdictions follow the World Meteorological Organisation guidelines for estimating the PMP
(WMO, 2009). The PMP is derived using observed "high efficiency" storms matched to a representative dew point
temperature. The moisture (i.e., rainfall) in the storm is then maximised by assuming the same storm could occur with
moisture equivalent to the maximum (persisting) dew point observed at that site.



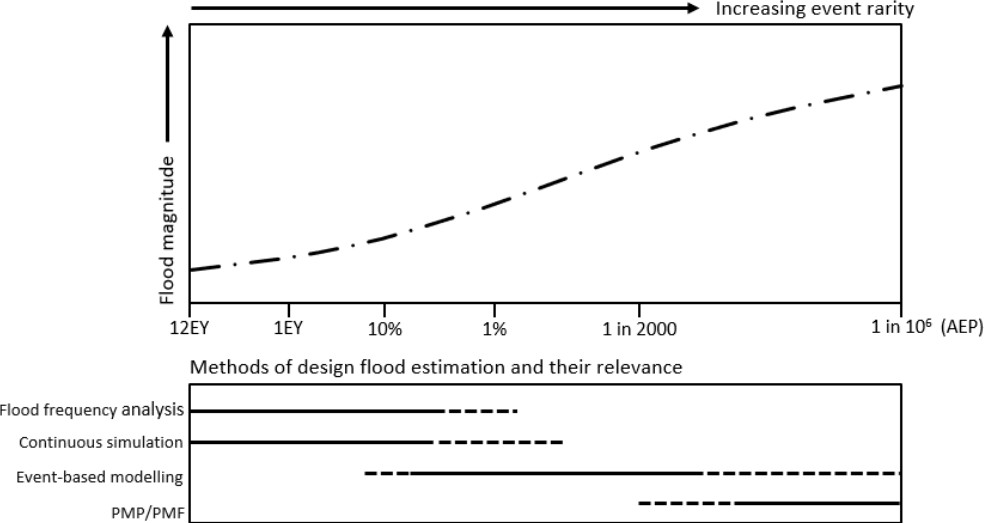


**Figure 1.** The relevance of different flood estimation approaches as a function of AEP. The top panel presents a typical flood frequency curve where the flood magnitude increases with event rarity. The bottom panel shows the range of AEPs for which various flood estimation approaches show efficacy. Dashed lines represent lower efficacy while solid lines represent the higher efficacy. Figure adapted from James Ball et al. (2019). The PMP is used an input in event-based models to derive the PMF.

The method adopted for design flood estimation depends on the problem being solved, the level of risk being designed for, and the available data. Most commonly, approaches based on event-based modelling are applied because data rarely exists at the location of interest, and if it does, it is often confounded by catchment non-stationary (e.g., urbanization, deforestation), or the record lengths are much shorter than the design AEP required.

3. **Methodology**

Systematic reviews represent a reproducible methodology for apprising the literature in the context of a specific topic or issue (Page et al., 2021). Reviews were undertaken for each of the three key flood estimation methods (flood frequency analysis, continuous simulation, and event-based modelling). To balance consistency between section authors and selection bias, each review section was assigned a lead author who was tasked with collecting scholarly articles from Scopus, with a secondary author tasked with reviewing the results of the systematic review. Articles were selected from 2011 onwards to ensure a broad coverage of evidence while ensuring that evidence is relatively contemporary. The literature search for each method of (or input to) design flood estimation contained different relevant keywords (see Supplementary Information for key words for each section). To limit the scope of the review geographically, searches were made for literature where either the title, abstract or keywords contained "Australia." To constrain the review only to climate change, literature was also required to contain "change" in either the title, abstract or keywords (it was deemed that using "climate change" would be too restrictive). These criteria represent



the foundation of the review, and the publication base was further supplemented by other sources of information, particularly in cases where specific terminology was used (e.g., the term "Clausius-Clapeyron" in the context of extreme rainfall) or where knowledge existed of additional publications or international research not identified through the keyword searches. We note here that the impact of sea level rise was excluded from the requirements of the systematic review as it is not explicitly part of the Australia Rainfall and Runoff guidance related to climate change (Bates et al., 2019).

To select relevant literature from the search results, articles were first filtered to remove duplicates. Following this, irrelevant articles based on a review of the abstracts, and then of the manuscript itself, were excluded. While the search terms aided inclusion in the systematic review, many studies were not relevant to the assessment of flood risk and were omitted. Finally, some additional studies (in particular, syntheses) were included based on the authors' knowledge of the literature. Details of the searches and the full list of articles reviewed is provided in the Supplementary Information.

Recognising the importance of IFD estimates in design flood estimation, and the large volume of available literature providing quantitative estimates of changes in extreme rainfall, an analysis was performed to understand the average effect size (change in extreme rainfall) and uncertainty intervals associated with this extreme rainfall. The analysis was inspired by meta-analysis techniques which quantitatively combine results from multiple studies (Field and Gillett, 2010) and used structured expert-elicitation methods consistent with those used by the IPCC (Zommers et al., 2020) in the following approach:

1. Where possible extreme rainfall change was quantified per degree of global temperature change (i.e., the global mean, including ocean and land regions), with variation to storm duration, severity (i.e., AEP), and location preserved. Global mean temperature was chosen to ensure consistency with the IPCC projections and to be representative of the climatic drivers of changes in moisture sources. The exception was for rainfall-temperature scaling studies, which use a local temperature as a proxy for anthropogenic climate change.

2. Assessment was made, through consensus between authors, whether there was enough evidence to calculate the average effect size with varying storm duration, severity, and location – and what, if any, distinction was to be made for these factors.

3. Co-authors independently used the collected evidence to determine their best estimate of the change in extreme rainfall as well as a likely range. Typically, each study weighted by how confident each author was in the evidence presented in the study. This included consideration of the study methodology (e.g., observation-based studies, model-based studies) and various statistical considerations (e.g., sample size and/or representativeness over the spatial domain).

4. Each of the best estimates from each author were then compared, and through a consensus process, a single central estimate was derived together with a likely (66%) range to represent assessment uncertainty.



## 4. Synthesis of the literature and systematic review

In this section, the literature is reviewed for each of the three key flood estimation methods (flood frequency analysis, continuous simulation, and event-based modelling). An overview of the implications of climate change on each method is first presented, followed by a systematic review using the keywords provided in the Supplementary Information. In the context of event-based (IFD) modelling each of the inputs to the design flood estimate are reviewed. For extreme rainfalls, the systematic review is followed by the results of the meta-analysis.

### 4.1 Flood frequency analysis
#### 4.1.1 Impact of climate change

Flood frequency analysis (or regional flood frequency analysis) generally uses annual maxima or threshold excess values of instantaneous flood data to derive a frequency curve by fitting an appropriate statistical model. Changes in flood maxima due to climate change are generally related back to changes in extreme precipitation. As temperature increases, so does the saturation water vapour of the atmosphere, leading to, all other things being equal, greater extreme precipitation, and hence pluvial flooding. However, flooding is dependent on the flood generating mechanism (Villarini and Wasko, 2021). In the absence of snowmelt, changes in antecedent soil moisture have been shown to modulate more frequent flooding while having a lesser impact on rarer floods, which are modulated by changes in extreme rainfall (Ivancic and Shaw, 2015; Wasko and Nathan, 2019; Neri et al., 2019; Bennett et al., 2018). Where snow is present, warmer temperatures cause a reduction in the frequency of rain-on-snow flood events at lower elevations due to snowpack declines, whereas at higher elevations rain-on-snow events become more frequent due to a shift from snowfall to rain (Musselman et al., 2018).

Across Australia, for frequent flood events in the order of annual maxima, more streamflow gauges show decreases in annual maxima than increases (Ishak et al., 2013; Zhang et al., 2016). There is a clear regional pattern, with decreases more likely in the extra-tropics, and increases more likely in the tropics. These changes have a strong correlation to changes in antecedent soil moisture and mean rainfall due to the expansion of the tropics (Wasko et al., 2021c; Wasko and Nathan, 2019). However, there is a statistically significant increasing trend in the frequency of rarer floods since the late 19[th] century (Power and Callaghan, 2016) due to increases in extreme rainfall (Wasko and Nathan, 2019; Guerreiro et al., 2018). Where research examines changes in flood frequency for Australia, it is often related to changes in catchment conditions (Kemp et al., 2020) or interannual variability (McMahon and Kiem, 2018; Franks and Kuczera, 2002). Specifically related to climate change, most studies for Australia argue trends in annual maxima have implications for non-stationary flood frequency analysis (Ishak et al., 2014), but often fail to detect statistically significant trends (Ishak et al., 2013; Zhang et al., 2016) due to natural variability (Villarini and Wasko, 2021).

In a review of the projection of flooding with warmer temperatures, Wasko (2021) summarised the global literature on non-stationary flood frequency analysis. It was noted that non-stationary flood frequency analysis for climate change is typically performed using time-dependent parameters (e.g. Salas et al., 2018). Wasko (2021) also noted that one of the shortcomings of non-stationary flood frequency analysis using a time covariate is the inability to project with confidence for climate change due to the lack of a causal relationship (see for example Faulkner et al. 2020).



Hence it is argued that any non-stationary flood frequency analysis should ensure that the statistical model structure
is representative of the processes controlling flooding (Schlef et al., 2018; Tramblay et al., 2014; Kim and Villarini,
2023; Villarini and Wasko, 2021), with a framework for model construction provided in Schlef et al. (2018). Examples
of physically motivated non-stationary frequency analysis from the global literature include using combinations of
rainfall, potential evaporation, soil moisture, temperature, and large-scale drivers of moisture transport as covariates
(Guo et al., 2023; Han et al., 2022; Tramblay et al., 2014; Schlef et al., 2018; Condon et al., 2015; Kim and Villarini,
2023; Towler et al., 2010). In principle, this is similar to studies performed in the United States, which have used
precipitation and temperature as covariates for non-stationary flood frequency analysis (Condon et al., 2015; Towler
et al., 2010; Kim and Villarini, 2023). But even the use of physically-based covariates is problematic as the covariates
should capture the differing processes that affect rainfall changes (Schlef et al., 2018), while GCM simulations may
not capture local flood controls (Villarini et al., 2015). A final complication is that, even if flood processes are captured
by the covariates, these statistical associations may not remain constant with climate change (Chegwidden, Oriana et
al., 2020; Zhang et al., 2022; Wasko, 2022). Possibly for the above reasons, there is little formal guidance for how to
perform non-stationary flood frequency analysis. One of the most well-developed guidance documents on flood
frequency analysis –Bulletin 17C (England et al., 2019) – while acknowledging the potential impacts of climate
change on flood risk, does not explicitly give guidance for climate change, but instead refers the user to published
literature for non-stationary flood frequency (Salas and Obeysekera, 2014; Stedinger and Griffis, 2011), leaving the
door open for a variety of analyses based on "time-varying parameters or other appropriate techniques." But Ahmed
et al. (2023) note there is a dearth of guidance on how to considerer non-stationarity in regional flood quantile
estimation, arguing alongside other reviews (Zalnezhad et al., 2022) that further research is needed on the impacts of
climate change on flood frequency analysis.

### 4.1.2 Systematic review
For Australia, the systematic review only yielded one manuscript. Using 105 catchments across the east coast of
Australia, Han et al. (2022) fit a non-stationary regional flood frequency model using the covariates of catchment area,
mean annual rainfall, mean annual potential evaporation, and rainfall intensity with a duration of 24 hours for the
target return period/exceedance probability. The proposed method is effective in capturing the differing trends with
differing recurrence intervals, and projections are derived, with more sites having increases projected for rarer events
(1 in 20 AEP) than for frequent events (1 in 2 AEP).

### 4.2 Continuous simulation
### 4.2.1. Impact of climate change
Where streamflow data is not available, flood frequency curves can be derived from simulated streamflow using a
rainfall-runoff model driven by long sequences of rainfall and evapotranspiration. The process of deriving flood
frequency curves through continuous simulation often necessitates the use of a weather generator to stochastically
generate the model inputs due to the long record lengths required for flood frequency estimation. For future climate
conditions, these model input time series are generally derived through downscaling methods (Fowler et al., 2007;
Teutschbein and Seibert, 2012) where GCM outputs are bias corrected and downscaled to create realistic inputs for
hydrologic (rainfall-runoff) models to simulate streamflow and consequently to derive flood frequency estimates.



Examples of this include Norway's flood guidance (Lawrence and Hisdal, 2011) and eFLaG in the UK (Hannaford et
al., 2023), where the magnitude of a flow of a given exceedance probability is compared to a reference period to
provide climate adjustment factors.
While changes in the hydrologic cycle and mean rainfall are largely constrained by the availability of energy, extreme
rainfall changes are constrained by moisture availability (Allen and Ingram, 2002). For Australia, increases in pan
evaporation have been observed (Stephens et al., 2018b), while for rainfall, longer dry spells between weather events
are projected (Grose et al., 2020), with a shift from frontal rainfall to convective rainfall, particularly in the southern
parts of the continent (Pepler et al., 2021). Rainfall events are expected to have, on average, a shorter storm duration
(Wasko et al., 2021a) with greater peak rainfall (Visser et al., 2023), and slower movement (Kossin, 2018; Kahraman
et al., 2021). As a result, although the frequency of extreme rainfall events may decline, when they do occur, the
extreme rainfall from the event is projected to increase (Grose et al., 2020) – with greater increases expected for more
extreme events (Wasko et al., 2023). Hence, just accounting for mean or extreme rainfall changes in isolation is not
sufficient and changes to the entire rainfall time series are required with climate change.

### 4.2.2. Systematic review


In climate literature the term "downscaling" is an umbrella term describing the conversion of coarse-resolution climate
model outputs to catchment-scale relevant outputs. The systematic review focused on "downscaling" yielded three
relevant manuscripts. In addition to these, one set of reports from the Australian Bureau of Meteorology was included
(Assessment Reports). Using five GCMs from the Coupled Model Intercomparison Project Phase 5 (CMIP5) and eight
global hydrologic models, Gu et al. (2020) projected changes up to the 1 in 50 AEP flood using the ISI-MIP trend-
preserving bias correction method (Hempel et al., 2013). Frequent floods were projected to decrease across large parts
of Australia, with some increases in the tropics. These patterns are amplified for rarer floods and again show decreases
(or no change) projected across the southern part of the country. The Australian Bureau of Meteorology has published
a dataset consisting of four CMIP5 GCMs and four downscaling methods gridded across the entire continent (Wilson
et al., 2022; Peter et al., 2023). In contrast to Gu et al. (2020) using this data (Wilson et al., 2022; Peter et al., 2023)
as input to the AWRA-L daily water balance model (Frost et al., 2018) the annual maxima and 1 in 20 AEP flood
events were projected to increase across most of the continent (Assessment Reports).
Wasko et al. (2023) used the MRNBC and QME downscaling methods that were found to perform best for hydrologic
variables (Vogel et al., 2023) in 301 locally calibrated catchment rainfall-runoff models across the continent.
Decreases in frequent flooding up to the 1 in 5 AEP were projected across large parts of the continent, while for rarer
events, the flood magnitude was projected to increase across the northern and eastern coasts. Differences in the results
were attributed to (1) the use of rainfall-runoff models that were calibrated locally (i.e., different parameter set for
each catchment) to flood frequency quantiles, whereas AWRA-L is calibrated to match dynamics of daily streamflow
and satellite soil moisture and evapotranspiration across Australia simultaneously using a single set of parameters
(Frost et al., 2018), and (2) due to the different downscaling methods adopted (Wasko et al., 2023). Recent research
has shown that, for hydrological applications, multi-variate bias correction that considers cross-correlations among
variables, temporal auto-correlations, and biases at multiple time scales (daily to annual) performs the best (Vogel et





al., 2023; Zhan et al., 2022). Further, both the bias correction and rainfall-runoff model calibration should be evaluated

for the target statistics of interest (flood frequency in this case), while also ensuring they are representative of flood

processes to guarantee robustness under change (Krysanova et al., 2018). Finally, Zhan et al. (2022) and Sharma et al.

(2021), among others, note that the uncertainty and variability in climate projections, complexity in selecting data, as

well as data processing, all hamper the adoption of climate data in continuous simulation. Indeed, Dale (2021) argues

that one of the primary requirements for design flood estimation moving forward is "a standard, accepted approach

for deriving time series rainfall that is representative of future climatic conditions for continuous simulation

modelling".

### 4.3 Event-based (IFD) modelling
#### 4.3.1 Processes affecting changes in Australian extreme rainfall

Before discussing the various complementary sources of knowledge that can provide insight into how climate change

could influence rainfall extremes, we first review the processes influencing changes in extreme rainfall. The primary

driver of extreme rainfall increase is the thermodynamic impact, a 6-7%/°C increase in the saturation vapor pressure

of the atmosphere, as dictated by the Clausius-Clapeyron (CC) relationship (Trenberth et al., 2003). Factors beyond

the thermodynamic impact have been discussed in various reviews and commentaries (Fowler et al., 2021; Allen and

Ingram, 2002; Pendergrass, 2018) and are summarised here. In general, for shorter duration rainfalls, the vertical lapse

rate (i.e., atmospheric stability) can affect the rate of rainfall. Atmospheric stability increases and rates of rainfall

decrease as temperature increases and the cloud base is lifted assuming moisture is unchanging. But if the moisture

increases, then the opposite is true, with rain more easily triggered. In addition, there can be an increase in buoyancy

creating stronger updrafts and deeper convection (referred to as super-CC scaling). Finally, dynamical drivers related

to changes in the global circulation can act to change the occurrence of rainfall extremes by changing storm tracks

and speeds, both amplifying and dampening the thermodynamic influence on rainfall extremes (Emori and Brown,

2005; Pfahl et al., 2017; Chan et al., 2023).

For Australia, extreme rainfall is typically associated with thunderstorms, cyclones, troughs or fronts (Dowdy and

Catto, 2017; Pepler et al., 2021; Warren et al., 2021), including tropical cyclones (TCs) in northern Australia (Dare et

al., 2012; Lavender and Abbs, 2013; Villarini and Denniston, 2016; Bell et al., 2019), east coast lows (ECLs) in the

east and southeast of Australia (Pepler and Dowdy, 2022; Dowdy et al., 2019) and thunderstorms (convective systems)

throughout Australia (Dowdy, 2020). Other physical processes leading to extreme rainfall occurrence include

enhanced advection of moisture to a region, such as from atmospheric rivers – large narrow bands of water vapor (Wu

et al., 2020; Reid et al., 2021; Black et al., 2021), and the temporal compounding of hazards such as heatwaves

impacting heavy rainfall occurrence (Sauter et al., 2023).

Tropical cyclones (TCs) can impact on northern regions of Australia, particularly in near-coastal locations, with their

occurrence generally from November to April (Chand et al., 2019). Although there is considerable interannual

variability in the number of TCs that occur near Australia, including influences of large-scale drivers such as the El

Niño-Southern Oscillation (ENSO), a significant downward trend in the frequency of observed Australian TCs has

occurred in recent decades (Dowdy, 2014; Chand et al., 2019, 2022). Climate models also indicate that TC numbers


in the Australian region are likely to continue decreasing in coming decades due to anthropogenic climate change
(Walsh et al., 2016; Bell et al., 2019; Bhatia et al., 2018; CSIRO and Bureau of Meteorology, 2015). However,
although fewer TCs are likely in a warmer world in general, this is more likely for non-severe TCs than severe TCs,
with extreme rainfall from TCs likely to increase in intensity at rates that could exceed 6-7%/°C of warming (Walsh
et al., 2016; Bhatia et al., 2018; Lighthill et al., 1993; Holland and Bruyère, 2014; Sobel et al., 2016; Emanuel, 2017;
Parker et al., 2018; Patricola and Wehner, 2018; Wehner et al., 2018; Knutson et al., 2020, 2019; Vecchi et al., 2019;
Kossin et al., 2020; Seneviratne et al., 2023). In addition to the frequency and severity, some studies have indicated a
potential poleward shift of TCs (Kossin et al., 2014), but there are considerable uncertainties around whether or not
this is occurring (Knutson et al., 2019; Bell et al., 2019; Chand et al., 2019; Tauvale and Tsuboki, 2019). Finally, some
studies have suggested a potential trend in the translational speed of TCs in a warming world (Kossin, 2018), while
others have suggested this might not be a significant change (Lanzante, 2019; Moon et al., 2019; Yamaguchi et al.,
349    2020).

East coast lows (ECLs) are cyclones near southeastern Australia that can be caused by both mid-latitude and tropical
influences over a range of levels in the atmosphere. Fewer ECLs are likely to occur due to anthropogenic climate
change, at a rate of about -10%/°C of global warming, with this change more likely for cooler months (Dowdy et al.,
2019; Pepler and Dowdy, 2022; Cavicchia et al., 2020). A recent study using RCM projections reported that the
number of cyclones exceeding the current 95[th] percentile for maximum rain rate is expected to increase by more than
25%/K in Australia's eastern seaboard and Tasmania under a high emissions pathway (RCP8.5) by 2070–2099. Both
the eastern seaboard and Tasmania are projected to have twice as many cyclones with heavy localised rain as in 1980–
2009 (Pepler and Dowdy, 2022). That study also found that about 90% of model simulations had at least one ECL in
the period 2070–2099, with a higher maximum rain rate than any in the period 1980–2009 for southeast Australia and
similarly for Tasmania. It is noted here that RCM projections are not at fine-enough scales to be convection-permitting
and so may not necessarily capture some changes in rainfall efficiency associated with enhanced convective processes
from increased atmospheric moisture capacity.
Convective storms, such as severe thunderstorms, can cause relatively localised storms as well as mesoscale
convective and linear systems (Hitchcock et al., 2021). As climate models have a limited ability to simulate fine-scale
aspects associated with thunderstorms (e.g., Bergemann et al. 2022), projections are typically based on environmental
conditions conducive to thunderstorm formation, such as convective available potential energy or other related
atmospheric metrics associated with deep and moist convection. Projections using environmental conditions such as
these have indicated a broad range of plausible changes in the frequency of thunderstorm environments for regions
throughout Australia, including potential increases or decreases depending on the metric or model selections used
(Allen et al., 2014; Brown and Dowdy, 2021). Some of the latest set of GCMs indicate an increase in convection-
related extreme rainfall over Australia relating to the Madden-Julian Oscillation (Liang et al., 2022).
Using lightning observations as a proxy for convective storm occurrence, a decline in the number of thunderstorms
during the cooler months of the year has been observed in parts of southern Australia, (Bates et al., 2015). Another



study based on rainfall observations and reanalysis data reported a trend since 1979 towards fewer thunderstorms for
most regions of Australia, with the strongest and most significant trends in northern and central Australia during the
spring and summer, in addition to increasing trends in thunderstorm frequency on the eastern seaboard (Dowdy, 2020).
However, the total rainfall associated with thunderstorms increased in most regions over the same time period, such
that the intensity of rainfall per thunderstorm increased at about 2-3 times the Clausius-Clapeyron rate (Dowdy, 2020).
Importantly, most of southern Australia saw an increase in the frequency of thunderstorms associated with rainfall of
at least 10 mm over the same period, particularly during the warm months (Pepler et al., 2021). That increase in rainfall
intensity exceeding the Clausius Clapeyron rate is broadly similar to some other studies based on observations and
modelling for Australia, including those focussed on short-duration extremes (Westra and Sisson, 2011; Bao et al.,
2017; Guerreiro et al., 2018; Ayat et al., 2022), with the larger increases tending to be in northern rather than in
southern regions. These high rates of change in rainfall intensity can occur from changes in rainfall efficiency, which
increases due to additional moisture capacity in a warmer atmosphere providing additional latent heat from
condensation as energy in the convective processes – so-called super-CC scaling. This process is relevant for
thunderstorms and TCs given the convective processes that provide energy for their formation and intensification, as
well as ECLs that sometimes have mesoscale convective features embedded within their broader synoptic structure
(Holland et al., 1987; Mills et al., 2010; Dowdy et al., 2019).
Extratropical cyclones and fronts can also sometimes cause extreme rainfall in southern Australia. Recent studies have
reported a trend towards fewer of these events, particularly during the cooler months of the year, including a reduction
in the frequency of events that generate at least 10 mm of rainfall (Pepler et al., 2021). Projections of extratropical
cyclones and fronts in this storm-track region of the Southern Hemisphere are broadly similar to the observed trends,
with studies indicating a general reduction in frequency for this region, particularly during the cooler months of the
year (Seneviratne et al., 2023; CSIRO and Bureau of Meteorology, 2015). The projections are also consistent with
observed reductions in multi-day rainfall events (Fu et al., 2023; Dey et al., 2019), which tend to be associated with
long-lived synoptic systems (i.e., at least 24 hours) such as extratropical cyclones.
Finally, the frequency of atmospheric rivers in Australia increased over the 1979-2019 period in one study (Reid et
al., 2022), and may increase in frequency in a warming climate, including near eastern Australia (Wang et al., 2023).
For example, a recent study demonstrated how an atmospheric river contributed to extreme multiday rainfall and
flooding in Sydney in March 2021, finding that, depending on the emission scenario, this type of atmospheric river
could increase in frequency by about 50-100% around the end of this century (Reid et al., 2021), but projections have
not been assessed in detail for elsewhere in Australia.
In summary, more intense rainfall extremes associated with TCs are likely to occur for northern Australia during the
warmer months of the year. For eastern Australia, fewer ECLs are likely to occur, but with an increase in the
occurrence of ECLs that cause extreme precipitation. For southern Australia, fewer extratropical cyclones and fronts
are likely to occur during the cooler months of the year, leading to a potential reduction in rainfall extremes during
these months. Increases in moisture transport by atmospheric rivers has also been reported, with the frequency of





strong atmospheric rivers potentially increasing by 50-100% in eastern Australia towards the end of this century. The
increased water vapour capacity of the atmosphere in a warming world can increase rainfall efficiency in some cases,
such as through enhanced latent heat from condensation contributing energy to the convective processes. This can
lead to increases in the intensity of extreme rainfall that are notably larger in magnitude than the 6-7%/°C increase
associated with the Clausius Clapeyron relation. Studies have indicated that increased rainfall efficiency in the order
of two or more times the Clausius Clapeyron relationship rate are plausible for short-duration rainfall extremes in
general for Australia (Guerreiro et al., 2018; Dowdy, 2020; Ayat et al., 2022).

### 4.3.2    Rainfall intensity
#### 4.3.1.1 Impact of climate change

IFD curves are typically derived using statistical models, such as the Generalized Extreme Value (GEV) distribution,
fitted to annual maximum rainfall across a range of durations to severities (AEPs). Anthropogenic changes in extreme
rainfall, both in their intensity and frequency, will therefore lead to changes in IFDs (Milly et al., 2008). In the
scientific literature, changes in extreme rainfalls are generally modelled using non-stationary frequency analysis with
appropriate covariates. While this is an active area of research (Schlef et al., 2023; Wasko, 2021) it has the same
shortcomings as non-stationary flood frequency analysis. Most studies use a time covariate to impart a temporal trend
(Schlef et al., 2023). However, there is evidence that accounting for the different drivers of extreme rainfall, for
example temperature for short duration rainfall, and climate modes such as the El Niño-Southern Oscillation (ENSO)
and the Indian Ocean Dipole (IOD) for long duration rainfall, can improve model performance (Agilan and
Umamahesh, 2015, 2017). This is consistent with the arguments put forward by Schlef et al. (2018) that covariates
should capture the thermodynamic and dynamic processes that affect rainfall changes. For non-stationary frequency
analysis, there is evidence emerging that GEV models should consider changes in both location and scale parameters
(Prosdocimi and Kjeldsen, 2021; Jayaweera et al., 2023). Finally, Schlef et al. (2023) summarised that for non-
stationary IFD analysis "the majority of covariate-based studies focus on the historical period, effectively reducing
the study to a sophisticated check for non-stationarity, rather than a framework for projection of non-stationary IDF
curves" and hence their application to the future period remains untested (Schlef et al., 2023).
Likely due to these difficulties in fitting non-stationary IFDs, the majority of climate change guidance for practitioners
is to scale the IFD rainfall depth or intensity using a climate adjustment (or uplift) factor derived from an assessment
of how extreme rainfalls are likely to change under climate change (Wasko et al., 2021b). Studies that assess potential
changes in extreme rainfall can be roughly separated into three categories: (1) studies that assess historical trends; (2)
studies that investigate the association of extreme rainfalls and temperature; and (3) studies that directly project
changes in extreme rainfall using model experiments.

#### 4.3.1.2 Systematic review

Our systematic review identified 40 manuscripts that quantified the relationship between temperature changes and
rainfall intensity, with the manuscripts roughly evenly split between the above three approaches. Projections were
almost always focussed on daily to multi-day rainfall extremes, with the exception of two studies that employed
regional models over small regions of Australia to provide projections of sub-daily rainfall (Mantegna et al., 2017;



Herath et al., 2016). In contrast, scaling studies were more likely to assess sub-daily rainfall, and about half the papers
assessing historical trends included sub-daily (usually hourly) rainfall.
Historical analysis of trends in high daily rainfall totals, such as the wettest day per year (Rx1D) or the 99th percentile
of the daily rainfall distribution, find a range of trends depending on the region and years used (Dey et al., 2019; Du
et al., 2019; Alexander and Arblaster, 2017; Sun et al., 2021; Liu et al., 2022a). Many older studies detected no
significant trend or a decreasing trend in Rx1D (e.g., Hajani and Rahman, 2018), including some large negative trends
when calculated for individual stations (Yilmaz and Perera, 2014; Chen et al., 2013). However, more recent studies
that draw on larger volumes of stations or gridded data more commonly detect increasing trends in Rx1D, many of
which are close to 7%/K (Wasko and Nathan, 2019; Dey et al., 2019; Guerreiro et al., 2018). Increases are most
apparent in the annual maximum intensity of events of no more than two days duration, which increased by between
13% and 30% over the period 1911-2016 for different regions of Australia (Dey et al., 2019). Changes in rainfall
intensity are less robust for longer duration rainfall events, with studies finding little change or even a decrease in the
intensity of the wettest five-day (Rx5D) period in southeast and southwestern Australia (Du et al., 2019; Fu et al.,
2023). Decreases in long-duration rainfall events are most evident during the autumn and winter (Zheng et al., 2015),
associated with extratropical weather systems (Pepler et al., 2020). While total rain days have decreased in many parts
of Australia, the intensity of rainfall on wet days may have increased (Contractor et al., 2018), as has the average
intensity of rainfall on days with thunderstorm activity (Dowdy, 2020).
There is increasingly strong evidence suggesting that an increase in the intensity of sub-daily rainfall has already
occurred. Guerreiro et al. (2018) found an average increase of 2.8 mm or 9.4% in the average wettest hour of the year
between 1966–1989 and 1990–2013 across Australia, equivalent to 19.5%/K, with increases observed at most stations
analysed. When divided into northern and southern Australia, trends were greater than 21%/K in the north, which has
seen a large increase in total rain over the same period (Dey et al., 2019); however, even in southern Australia,
increases were larger than those expected based on Clausius-Clapeyron for frequencies up to the seven wettest hours
(7EY) per year, and close to 14%/K for the wettest four hours per year. In Victoria, studies have found an 89% increase
in the frequency of hourly rainfall > 18 mm/h (Osburn et al., 2021) between 1958-1985 and 1987-2014, as well as
increases in hourly totals > 40 mm/h (Tolhurst et al., 2023). Yilmaz and Perera (2014) also found increasing trends in
Melbourne rainfall intensities for durations of three hours or less between 1925-2010, with 1 in 2 AEP values 5-7%
higher when calculated using data from 1967-2010 vs 1925-1966 (~13-17%/K), though not all differences were
statistically significant.  In southeast Queensland and northeast New South Wales, increasing trends for annual maxima
for events with a duration of less than 12 hours have been reported (Laz et al., 2014), while Chen et al. (2013) reported
that the heaviest rainfalls at timescales of six minutes to six hours increased between the earlier and later 20th century
by more than 20% in Melbourne, Sydney and Brisbane. Very large increases of ~20%/decade in sub-hourly rainfall
have also been identified in Sydney using both radar and rain gauge data based on the short period of 1999-2017 (Ayat
et al., 2022). Trends tend to be strongest for convective rainfall, which has its largest contribution to short duration
events and during the warm half of the year. For instance, heavy rainfall in Greater Sydney during the summer months
increased by more than 6%/decade for all durations from six minutes to 48 h over 1966-2012 (Zheng et al., 2015).



Scaling studies typically use quantile regression on rainfall-temperature pairs or linear regression on extreme rainfall
percentiles after grouping records by temperature classes to calculate the relationship between day-to-day temperature
variability and the upper tail of the rainfall distribution, as represented by the 90[th] or 99[th] percentile of rainfall at a
given temperature (Wasko and Sharma, 2014). While early scaling studies used dry bulb air temperature, such
approaches were sensitive to the cooling influence of rainfall on air temperature as well as the temporal and spatial
scales of rainfall (Bao et al., 2017; Barbero et al., 2017), and often found negative scaling in the northern tropics
(Wasko et al., 2018). More recent studies have found more homogenous results by scaling against moisture
availability, most commonly the dewpoint temperature, as well as accounting for intermittency in precipitation events
(Visser et al., 2021; Schleiss, 2018). Studies typically find a median scaling over Australia of 7-8%/K for daily rainfall
(Magan et al., 2020; Roderick et al., 2020; Bui et al., 2019; Wasko et al., 2018; Ali et al., 2021b; Visser et al., 2020).
This regional convergence to Clausius-Clapeyron scaling hides larger variability in the scaling at local station scales,
ranging typically between 5-10%/K, although in the northern tropics many stations exhibit scaling greater than 14%/K
between rainfall and dewpoint temperature (Magan et al., 2020; Wasko et al., 2018).
Scaling is typically stronger for sub-daily rainfall, with median scaling over Australia typically 8-10%/K and scaling
in tropical regions frequently exceeding 14%/K (Wasko et al., 2018; Ali et al., 2021b; Visser et al., 2021). For rarer
events, Wasko and Sharma (2017) used a stochastic weather generator conditioned on temperature and found hourly
rainfall scaling for Sydney and Brisbane increased from 6-9%/K for an AEP of 1 in 2 to 10-12%/K for a 1 in 10 AEP
and 18%/K for a 1 in 100 AEP, although the uncertainty ranges were large. Scaling rates exceeding 15%/K between
dewpoint temperature and daily rainfall over Australia have also been calculated using a global $0.25° \times 0.25°$
latitude/longitude resolution model (Zhang et al., 2019), although scaling in the Sydney region was ~4%/K for hourly
rainfall using a 2 km convection permitting model (Li et al., 2018).
GCMs are not expected to accurately simulate rainfall extremes due to deficiencies at representing the key phenomena
responsible for extreme rainfall including convection and thunderstorms or tropical cyclones. This is particularly true
of short-lived or sub-daily extremes, with GCMs better at simulating daily or longer extremes such as extratropical
lows, which cause widespread and prolonged heavy rainfall Kendon et al., 2017). Projections from CMIP5 models
between 1986-2005 and the late 21[st] century (~2081-2100) indicate an increase in RX1D under a high emissions
scenario (Alexander and Arblaster, 2017), with regional mean increases in RX1D ranging from 13% in Eastern
Australia to 19% in Northern Australia (~4-6%/K) (Climate Change in Australia). A 4%/K increase in RX1D was also
found by Chevuturi et al. (2018) when comparing a 2-degree warmer world with historical simulations, while Ju et al.
(2021) found an 11% increase in RX1D in a 2-degree warmer world (5.5%/K). Models in the Coupled Model
Intercomparison Project Phase 6 (CMIP6) simulate a slightly smaller change in RX1D, with a 6.2-7.3% increase in
Rx1D for Australia between the preindustrial climate and the 2-degree warming level and a 10.3-11.2% increase by 3
degrees (3-4%/K, Gutiérrez et al., 2021) and a 9.4% (~3%/K) increase in Rx1D by the end of the century (Grose et
al., 2020).
Results from regional climate models are broadly consistent with GCMs for daily rainfall, including a projected
regional mean increase of 5.7%/K  in the 99[th] percentile of wet days using the NARCliM ensemble (Bao et al., 2017)



and larger increases in the 99.5[th] (6.5%/K) and 99.9[th] (9.2%/K) percentiles. Pepler and Dowdy (2022) also found a
4%/K increase in the frequency of days exceeding the 99.7[th] percentile using a CMIP5-based RCM ensemble, with
the largest increases projected in Tasmania (12%/K), while Herold et al. (2021) reported a doubling in the frequency
of current 1 in 20 AEP events by 2060-2079. Projected increases are smaller for multi-day rainfall, with a median
increase in Rx5D of 10% (~3%/K) reported in Sillmann et al. (2013), 4%/K in Ju et al. (2021), and no significant
change in Chen et al. (2014). While fewer studies have assessed changes to less frequent rainfall extremes, these are
typically larger than the increases projected for annual maxima. For instance, CMIP5 models simulate a 22-26%
increase (7-8%/K) in the 1 in 20 AEP daily rainfall by the end of the 21[st] century (Climate Change in Australia), and
statistically downscaled climate data project a similar 20% increase in the 1 in 50 AEP by the end of the century
(6%/K; Wasko et al., 2023). Slightly smaller increases for the 1 in 10 AEP of 15.5% by the end of the century were
found using CMIP6 models (~5%/K, Grose et al., 2020).
Studies investigating the projection of sub-daily rainfall extremes are rare for Australia, but regional modelling for the
Tasmanian region indicated increases of greater than 40% in AEP of 1 in 10 and rarer in a 2.9-degree warmer world;
more than 14%/K (Mantegna et al., 2017). This is consistent with the stronger observed trends and scaling rates
reported for rainfall of short durations. Projected increases are likely to be larger for convective extremes, which
dominate sub-daily rainfall and are poorly simulated even in regional climate models. For example, Shields et al.,
(2016) projected a 12.5% increase in convective rain rates above the 95[th] percentile in the Australasian region using a
0.5° × 0.5° latitude/longitude global model by the late 21[st] century (~4%/K) but no change in large-scale rainfall.
Finally, regional model experiments also indicate increases of 15% in tropical cyclone rain rates per degree of SST
increase (Bruyère et al., 2019).
**4.3.1.2 Meta-analysis**
Where possible, observed or projected changes were extracted from each paper or dataset. Absolute changes were
converted to changes as a percent per degree of warming, with the global mean warming over the appropriate time
period extracted either from the Berkeley Earth Surface Temperature dataset (Rohde and Hausfather, 2020), or the
ensemble mean for the corresponding CMIP generation and emissions scenario. These quantitative results are
summarised in Figure 2, with extended details provided in the Supplementary Table. The centre changes are central
estimates changes in extreme rainfall amounts converted to %/K. The type of central estimate (median or mean) is
indicated in the Supplementary Table. Minimum and maximum changes are the largest range of changes reported by
each study; these are usually minima and maxima (for example across stations). It is noted that some papers are
included in Figure 2 multiple times for different durations and exceedance percentiles.



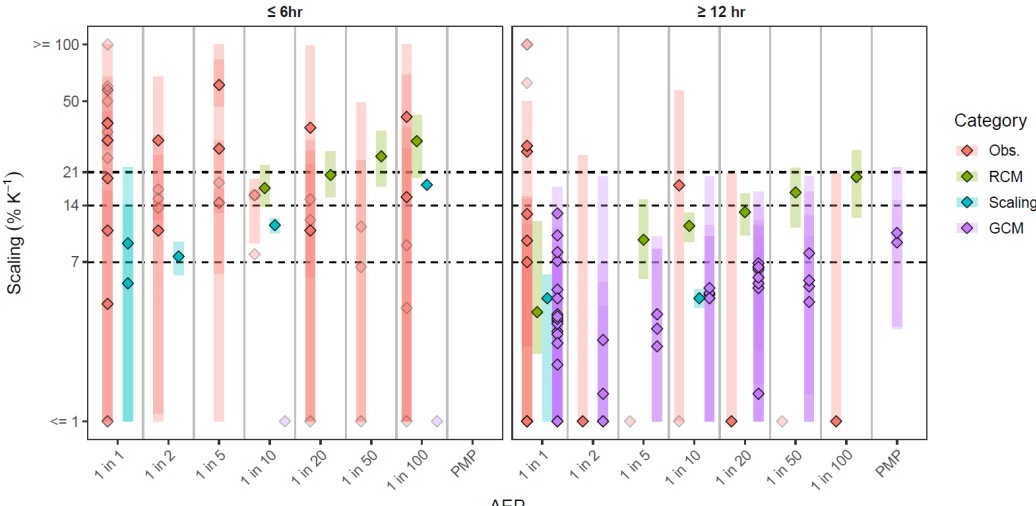

**Figure 2.** Summary of extreme rainfall change standardised, where possible, in per degrees of global temperature change. Note that rainfall-temperature scaling studies use local temperatures. Dashed lines indicate Clausius-Clapeyron (CC), 2xCC, and 3xCC scaling respectively. Diamonds indicate the central estimate of scaling and shaded bars indicate the range (where possible the minimum to maximum) of scaling estimates. Diamonds are opaque for results in which there was higher confidence and transparent for estimates in which authors found "disqualifying features" that significantly lowered weighting in the meta-analysis. The few studies with AEPs between the values shown here were included in the nearest AEP for this plot.

By consensus, it was deemed that the results for the meta-analysis would focus on daily and hourly rainfall durations as the majority of studies focus on these two durations. Additionally, the mechanisms that cause extreme rainfall at the two durations are often distinct (albeit noting that short duration extremes are often embedded in longer duration extremes). The potential for rates of change to vary both by location and exceedance probability was also explored. In relation to changes by location, it is well known that there is significant heterogeneity in the rainfall-generating mechanisms across the Australian landmass. However, when comparing the published scaling rates across the different geographies, there was insufficient evidence to quantify the differences between regions, with a relative scarcity of studies in regions outside of the populated areas of eastern Australia, and few consistent methodologies applied to all of Australia. Similarly, although there is some evidence that rarer extremes are likely increasing more than frequent extremes, it was deemed there was not enough evidence to quantify this difference through the meta-analysis (See Figure 2). This was because of (1) the large variability of extreme rainfall changes between studies relative to the variability with AEP, and (2) where there appears to be a trend with AEP this is generally a result of a single study analysing multiple AEPs. Hence for the proposed uncertainty intervals were developed to encompass much of the variability across space and by exceedance probability.





Multiple co-authors independently used the available evidence to determine their best estimates of central scaling rates
and the likely range of extreme rainfall change, for events rarer than the annual maxima up to the PMP. For both daily
and hourly durations, each relevant study was assigned a weight, where the weights across the studies summed to one.
The weights were assigned based on the type of evidence (I.e., trend, association, or projection), the study
methodology, the number of sites analysed, the age of the study and its spatial extent, and theoretical considerations.
These weights were then used to obtain a best estimate of the change in extreme rainfall. A consensus was drawn
between the participating co-authors with regard to the central (median) estimate and the likely range (66%) of extreme
rainfall change. The consensus scaling rates and ranges are shown in Table 1.
**Table 1.** Results of meta-analysis presenting extreme rainfall change, using a multiple-lines-of-evidence approach
that draws on the studies in the Supplementary Table. This synthesis is based on a review of all studies covering
extremes from the annual maxima through to the probable maximum precipitation (PMP) event (see Section 4.3.3
579         for further information on the PMP). The estimates are presented per degree global temperature change.

|  | <=1 hr | >1 hr and <24 hr | >=24 hr |
|---|---|---|---|
| Central (median) estimate | 15%/K | Interpolation zone | 8%/K |
| 'Likely' range (corresponding to ~66% range) | 7%-28%/K | Interpolation zone | 2%-15%/K |


Weightings given by individual authors reflected the following findings. At daily timescales, RCM projections and
scaling approaches typically had higher scaling rates than GCM projections, likely due to deficiencies in GCMs
representing key extreme rainfall generation processes. Moreover, many observational studies used few sites with
limited spatial coverage. In most studies using historical data across larger regions (global or Australia wide) and
recent periods, results were between 4-10%/K, with a central estimate of 8%/K for rarer events (e.g., 1 in 100 AEP),
noting also that a greater weight was given to those global and Australia-wide studies. The likely range encompasses
small but non-negative changes, which are most likely due to changes relevant to more frequent, multi-day events of
72+ hour duration. The likely range also encompasses potential scaling of at least twice the Clausius-Clapeyron rate,
most likely for rarer events such as 1 in 100 AEP and for locations in northern Australia.
For sub-daily timescales, estimates of change are predominantly based on historical observations (trends), due to a
relative paucity of projection information. These studies suggest that changes below the Clausius-Clapeyron rate of
7%/K are unlikely, with potential changes in excess of 15%/K observed for rarer events. This is broadly consistent
with the single available regional model study (Mantegna et al., 2017), which had projected increases of 16%/K for a
1 in 10 AEP and 29%/K for 1 in 100 AEP. Slightly weaker changes are found in scaling studies compared to the other
lines of evidence, with the tropics again showing evidence of greater increases compared to the south. The likely range
hence incorporates this spatial inhomogeneity noting that greater uncertainty exists on the upper estimate of change
than the lower estimate. While the meta-analysis central estimate of 15%/K is based on the best available information,





there is an urgent need for more detailed assessment of changes in sub-daily rainfall in a changing climate using
convection-permitting models.

### 4.3.3     Probable maximum precipitation

#### 4.3.3.1 Impact of climate change

The PMP is defined as the greatest depth of precipitation meteorologically possible under modern meteorological
conditions for a given duration occurring over a catchment area or a storm area of a given size, at a certain time of the
year (WMO, 2009). It needs to be recognised that this theoretical definition differs from its "operational estimate,"
which is based on a set of simplifying assumptions and calculated from an observational sample of
hydrometeorological extremes (Schaefer, 1994). Hence, in Australia and elsewhere, successive estimates of the PMP
adopted for design purposes have increased over time as methods and data sets change (Bureau of Meteorology, 2003).
As a result, changing PMP estimates for climate change is heavily dependent on the operational methods employed.
The methods used to derive operational PMP estimates can be broadly divided into statistical methods and
hydrometeorological methods. Statistical methods are commonly used in engineering studies as they can be applied
with little effort and do not require hydrometeorological expertise. The most widely used statistical approach was
developed by Hershfield (1965) and is based on enveloping the observations obtained from a large number of rainfall
gauges to extrapolate a simple 2-parameter (Gumbel) distribution. Hydrometeorological methods used to derive
operational estimates include approaches based on the maximisation of local storm data, referred to as in-situ
maximisation, the transposition of extreme storms nearby to the catchment with similar topography, known as storm
transposition, and the enveloping of storm data over a large region after adjusting for differing moisture availability
and topography, known as generalised methods. Generalised methods differ from the in-situ and transposition methods
in that they use all available data over a large region and include adjustments for moisture availability and differing
topographic effects on rainfall depth. Generalised PMP methods are employed in Australia as well as a number of
other countries, including New Zealand (Thompson and Tomlinson, 1995), India (Rakhecha and Kennedy, 1985),
China (Gu et al., 2022), and the USA (England et al., 2020). For Australia, the storm transposition zone varies with
climate region as the mechanisms driving extreme rainfall vary.
In generalised hydrometeorological methods, the PMP event is assumed to originate from the simultaneous occurrence
of a maximum amount of moisture (moisture maximisation) and a maximum conversion rate of moisture to
precipitation (storm efficiency). Moisture maximisation involves multiplying observed storm precipitation depths by
the ratio of the seasonal maximum precipitable water for the storm location to the representative precipitable water
for the storm, with the precipitable water estimated from surface dewpoint data assuming saturation and pseudo
adiabatic conditions. This assumes that in a large sample of storms recorded over a long period at least one storm
operates near maximum efficiency.
Potential increases in future daily PMP estimates are predominantly founded on projected increases in atmospheric
water vapor, which have been found to closely follow temperature changes with an approximate Clausius-Clapeyron
relationship of 7% per 1°C warming (noting that this does not consider potential changes in rainfall efficiency). While
the WMO manual (WMO, 2009) makes no allowance for long-term climatic trends, one of the most comprehensive





studies that examined changes in maximum water vapour concentrations across the globe found increases in
atmospheric water vapor of 20%–30% by the end of the century (Kunkel et al., 2013), approximately consistent with
the CC relationship. Kunkel et al. (2013) adopted a "hybrid" approach that merged traditional hydrometeorological
PMP methods with outputs from an ensemble of seven GCMs, an approach that is seen as an advance on traditional
PMP estimates as it incorporates simulated historical and future climate model data (Salas et al., 2020). They found
that the PMP will change by an amount comparable to the mean water vapour changes, with little evidence for changes
in storm efficiency (Kunkel et al., 2013); however it is noted that GCMs do not simulate many of the key process that
could lead to changes in storm efficiency. The relatively minor importance of changes in storm efficiency compared
to precipitable water under climate change was also found by Ben Alaya et al. (2020), who based their conclusions on
an analysis of non-stationarity in a bivariate model of precipitable water and storm efficiency using temperature as a
covariate.
Since Kunkel et al. (2013), many other hybrid approaches have been applied using either global or regional climate
models, and similar results have been found for catchment- or region-specific studies in northern America (Beauchamp
et al., 2013; Chen et al., 2017; Cyphers et al., 2022; Clavet-Gaumont et al., 2017; Rousseau et al., 2014; Rouhani and
Leconte, 2020; Labonté-Raymond et al., 2020), Chile (Lagos-Zúñiga and Vargas M., 2014), and Korea (Lee et al.,
2016). While one study projected decreases in the PMP using a hybrid modelling approach, it was based on a single
GCM model (CanESM2) and the projections were for a region in the southeast of the Caspian Sea (Afzali-Gorouh et
al., 2022). Other region-specific studies have applied physically-based approaches using regional atmospheric models
and found results that are consistent with the Clausius-Clapeyron relationship in north America (Ishida et al., 2018;
Gangrade et al., 2018; Rastogi et al., 2017), China (Liu et al., 2022b), and Chile (Lagos-Zúñiga and Vargas M., 2014).
Statistical methods based on Hershfield (1965) have also been used to assess the non-stationarity of PMP estimates,
where a recent study (Sarkar and Maity, 2020) used a global reanalysis data set to conclude that global PMP estimates
have increased by an average of 25% over the world between the periods of 1948-1977 and 1978-2012. These changes
are appreciably larger (e.g., about quadruple) than what would be expected from the Clausius-Clapeyron relationship,
though differences between statistical and hydrometeorological methods are evident in other studies in Canada
(Labonté-Raymond et al., 2020), India (Sarkar and Maity, 2020), Vietnam (Kawagoe and Sarukkalige, 2019) and the
USA (Lee and Singh, 2020). The degree of conservatism associated with the statistical method (i.e., the tendency to
produce high estimates) is heavily dependent on the robustness of the envelope curves. Given the lack of physical
reasoning in the statistical method, it is difficult to reconcile differences with estimates derived using
hydrometeorological concepts. This is also true of generalised methods, which in principle do not vary with storm
duration, with research into changes in the PMP with climate change largely using daily rainfall data.
**4.3.3.2 Systematic review**
A systematic search yielded one recent paper relevant to projected changes in operational PMP estimates for Australia
(Visser et al., 2022), with Salas et al. (2020) summarising existing methods and findings. Visser et al. (2022) undertook
an analysis of moisture availability, comprising dewpoint data from 30 synoptic stations across Australia covering the
period from 1960 to 2018 and 3-hourly ERA5 reanalysis data covering the period from 1979 to the present (Hersbach



et al., 2020). It was found that the annual maximum persisting dewpoints have increased leading to increased PMP
estimates. Projections of dewpoint temperature were used to derive future PMP estimates across Australia using the
ACCESS-CM2 model. The projected results showed increases of 4%-29% (average of 13%) by 2100 for SSP1-2.6
and 12-55% (average of 33%) for SPP5-8.5 (Visser et al., 2022). If global temperature increases are used, these
changes translate to average increases slightly greater than the Clausius-Clapeyron relationship (e.g., 8.9%/K for
SSP5-8.5).
Jakob et al. (2009) investigated how the local moisture availability, storm type, depth-duration-area curves and relative
storm efficiency used in deriving operational PMP estimates might be changing over time, and how the identified
changes have impacted the PMP estimates. The analysis was based on data from 38 locations across Australia from a
combination of upper-air (radiosonde) and surface dewpoint observations. No large-scale significant changes in
moisture availability were found, though significant increases were found along parts of the east coast, as well as a
region in south-eastern Australia with summer decreases. When comparing moisture availability for a historical
climate period (1981-2000) and the next few decades using outputs from a single global climate model, they found
the 90th percentile values increased from the 2020s to the 2050s and the 2090s, however they also found some evidence
for lower extreme moisture availability in some regions. Similar to the above studies, they found little evidence for
significant changes in storm efficiency, depth-duration-area curves, or storm types, and no significant changes were
found in generalised rainfall depths (again noting that such global models are not expected to simulate some of the
key rainfall generation processes). The results obtained by Jakob et al. (2009) are not inconsistent with those of Visser
et al. (2022), but the difference in conclusions may be explained by the longer and more extensive data sets used by
Visser et al. (2022) and the updated global climate model outputs used to project the dewpoint temperatures.
Despite this compelling evidence, there is no formal recommendation for increases in PMP estimates with the Manual
on Estimation of Probable Maximum Precipitation (WMO, 2009) in their chapter on "PMP and Climate Change"
summarising the results of Jakob et al. (2009). To the best of the authors' knowledge, no agency responsible for
providing operational PMP estimates for design purposes anywhere in the world has yet provided uplift factors to
ensure that the PMP estimates based on historic observations are relevant to future conditions, despite the majority of
studies into impact of climate change on the PMP finding the PMP is likely to be increasing at the CC rate for daily
rainfall.

### 4.3.4 Temporal and spatial patterns
#### 4.3.4.1 Impact of climate change

The temporal and spatial patterns of extreme rainfall have long been recognised as important factors in determining
the magnitude of a flood event (Herrera et al., 2023). Conceptually, as weather systems change and storms intensify
due to increases in temperature, changes in both the temporal and spatial pattern of rainfall are expected with
anthropogenic climate change. Given that sub-daily rainfalls are expected to intensify more than daily rainfall (Section
4.2.1) this implies that storm temporal patterns will also intensify. In the design flood paradigm, once a rainfall depth
has been estimated from the appropriate IFD relationship, a temporal profile is used to distribute the total rainfall
across the storm duration. When the rainfall distribution across the storm duration is less uniform, higher flood peaks


will generally occur (Ball, 1994). For example, front or rear loaded storms, where more than 50% of the total rainfall
falls in either the first half or the second half of the storm respectively (Visser et al., 2023), can have differing impacts
on flood peaks through their interactions with any storage (natural or constructed) in the catchment.
In the context of design flood estimation, as the underlying data for the IFD relationships is point rainfall, the influence
of spatial scale on average rainfall intensities is considered through ARFs. For small catchments the point rainfall
provides a reasonable approximation of the catchment average rainfall. However, for larger catchments, it is less likely
that the most intense rainfall in a storm will occur over the whole catchment and the catchment average rainfall for
any particular event will be lower than the point rainfall represented by the IFD relationship. ARFs represent this
expected rainfall reduction, with the reduction dependent on the catchment area, storm duration and frequency.
**4.3.4.2 Systematic review**
Some limited research has been undertaken with respect to changes to temporal patterns and spatial patterns of design
rainfalls, primarily using scaling relationships calculated from observed data, while there exists some limited
modelling via dynamic downscaling for the Sydney region. A total of seven papers were found as part of the systematic
review. The findings to date suggest that temporal patterns are becoming more front-loaded (greater percentage of
precipitation falling earlier in the storm) with higher temperatures. There is also an increase in the proportion of rain
falling in the wettest period of the storm, leading to increased peakiness (less uniformity) of the temporal patterns.
Temporal pattern changes have been analysed in two main ways. The first is broadly based on the average variability
method, whereby the changes in the proportion of rainfall within a period are calculated. For example, Wasko and
Sharma (2015a) found for 1 hour storm bursts, the highest 12-minute period had a median scaling of 2.1% per degree
temperature increase for Australia. The scaling rate was dependent on the duration of the storm and the latitude of the
station. Wasko and Sharma (2015b) identified 500 one-hour bursts for five stations, stratified them into five
temperature bins and calculated the temporal pattern using the average variability method for each bin. In general, the
highest temperature bin had peakier (i.e., less uniform) temporal patterns than the lowest temperature bin. Wasko and
Sharma (2017) also used the average variability method to calculate the scaling of temporal patterns. These later
analyses were based on first fitting a stochastic rainfall generation model to historical observations, and then using
regression models to explore the relationships between the rainfall generation model parameters and temperature. For
simulations representing the end of the 21$^{st}$ century under RCP8.5, the peak rainfall fraction in the temporal patterns
increased from 40% to 50% for two models that were fitted separately for Brisbane and Sydney.
Australia's flood guidance (Ball et al., 2019a) has moved away from using the average variability method for temporal
patterns, and instead now provides an ensemble of temporal patterns for design rainfall analyses. Consistent with this
approach, Visser et al. (2023) provide the most comprehensive analyses of scaling relationships for temporal patterns
for Australia. From an original database of 1489 rainfall gauges 151 stations had sufficient data for scaling analysis,
and trends could be calculated for 55 locations from 1960-2016, with 28 stations having coincident temperature and
precipitation data. It was found that storms have tended to become more front-loaded, with storms also tending to
become more front-loaded when the coincident temperature was higher. There is a strong regional pattern in the
proportion of front-loaded events, ranging from 50% of events in the south of Australia to close to 70% of events in



the tropics. Scaling relationships for the temporal patterns were found to be stronger when related to temperature
rather than dew point temperature.
The only study to directly calculate ARFs in the context of climate change is Li et al., (2015). In this work, ARFs were
calculated for the Sydney region using a high-resolution RCM. It was found that for 1hr storms ARFs would increase
(i.e., larger future storms) whilst for longer durations (6 to 72 hr) ARFs would decrease, with the largest decreases for
large catchment areas and the rarest events. But as this analysis was based on a single climate model applied over a
limited geographical domain it is not possible to generalise these results. Calculating ARFs from the RCM also
assumed that the point rainfall to 4 $km^2$ ARF would not change in the future (as 4 $km^2$ was the resolution of the RCM
so smaller area ARFs could not be calculated).
Other studies have analysed changes to spatial patterns of storms, but further work will be required to relate their
findings to methods such as ARFs used with design rainfalls. Wasko et al. (2016) found that the effective radius of
storms decreased with temperature at over 80% of the stations analysed in Australia using quantile regression for
storms above the 90[th] percentile. For stations classified as temperate, this decrease in effective radius was despite an
increase in peak precipitation, which suggested that moisture was being redistributed from the edge of the storms to
the centre. Li et al. (2018) reproduced these results for the Sydney region using RCM simulations. However, in both
studies the storms were limited to radii of 50 km and were assumed to be circular. Li et al. (2018) pointed out that
there were good opportunities to use RCM simulations to analyse changes in storm advection and not limiting the
analyses to circular storms.
Finally, Han et al. (2020) used copulas to analyse the spatial dependence of monthly maximum rainfalls. They found
that around 40% of the stations had decreasing trends in connectivity and that the overall average connectivity was
lower for storms associated with higher dewpoint temperatures, particularly in southern Australia. However, the
analyses were not seasonally stratified and therefore it is not clear if the findings could also be explained by the
seasonally different rainfall mechanisms. Although evidence is emerging for temporal and spatial clustering of storm
events both in Australia and globally (e.g., Chan et al., 2023; Chang et al., 2016; Ghanghas et al., 2023; Kahraman et
al., 2021; Tan and Shao, 2017), the evidence for changes in the spatial pattern of precipitation, compared to changes
in the temporal pattern of precipitation, remains weaker.
### 4.3.5    Antecedent wetness
#### 4.3.5.1 Impact of climate change
When rainfall falls on a catchment, there a range of different runoff processes that lead to catchment runoff and
subsequent streamflow. These runoff processes include infiltration excess or Hortonian overland flow, saturation
excess runoff, variable source area, partial area runoff, subsurface storm flow, and impervious area runoff. In
modelling these runoff processes in design flood estimation, the rainfall is partitioned into direct flow or runoff, which,
along with baseflow, contributes to the observed flood hydrograph, and rainfall losses that do not influence the flood
event's hydrograph. Rainfall losses primarily result from: 1) interception by vegetation and man-made surfaces which
are eventually evaporated 2) depression storage on the land surface ranging in size from soil-particle-sized depressions



to lakes; and 3) infiltrated water stored in the soil, which may later contribute to baseflow (Hill and Thomson, 2019;
Pilgrim and Cordery, 1993; O'Shea et al., 2021).
Physically, rainfall losses are largely influenced by antecedent soil moisture and soil properties, which govern the
hydraulic gradient of the soil and thus affect the rate of infiltration (Liu et al., 2011; Bennett et al., 2018). Antecedent
soil moisture is a strong modulator of the flood response (Tramblay et al., 2010; Pathiraja et al., 2012; Woldemeskel
and Sharma, 2016; Wasko et al., 2020; Brocca et al., 2009; Quintero et al., 2022) and is influenced by variability at
multi-annual and multi-decadal time scales (Kiem and Verdon-Kidd, 2013). Incorporating information regarding
antecedent soil moisture into loss models has also been shown to improve flood estimates (Cordery, 1970; Tramblay
et al., 2010; Sunwoo and Choi, 2017; Bahramian et al., 2023); these loss models have been incorporated into the
Australia's flood guidance (Hill et al., 2016).
To model the flood response in event-based flood routing models, it is necessary to conceptualise rainfall losses and
employ a mathematically explicit representation. More complex loss models, such as Horton's method, conceptualise
the infiltration as decreasing exponentially as the soil saturates, whereas the Green-Ampt method assumes a sharp
wetting front exists in the soil column, separating a saturated upper soil layer from the underlying soil layer that
contains some initial moisture content (Rossman, 2010). Research has also explored the merits of hybrid methods
where continuous simulation is used to condition the initial state of the catchment before modelling the discrete flood
event using an event-based flood model (Heneker et al., 2003; Sheikh et al., 2009; Li et al., 2014; Yu et al., 2019;
Stephens et al., 2018a). Despite authors arguing that loss models should involve modelling physical representations
of the runoff process (Kemp and Daniell, 2016), there has been limited adoption in practice of more complex
approaches to loss modelling (Paquet et al., 2013). This is because the benefits of estimating rainfall losses relevant
to floods using physical process-based models are limited due to their complexity and incomplete understanding of
runoff generation processes as well as the inadequate availability of hydrological data (Pilgrim and Cordery, 1993).
For example, complex fully-distributed models often seek to resolve processes at spatial and temporal scales for which
data is limited or unavailable, and consequently such models are more liable to overfitting, leading to poor predictive
capabilities. As a result, parsimonious lumped models of rainfall loss are commonly employed.
Amongst the most used parsimonious lumped models of rainfall loss are the initial loss continuing loss model (ILCL),
the Probability Distributed Model (PDM), the Soil Conservation Service Curve Number (SCS-CN) and the initial loss
proportional loss (ILPL) model (Pilgrim and Cordery, 1993; O'Shea et al., 2021; US Army Cops of Engineers, 2000).
Broadly, these models divide losses into an initial loss, whereby all rainfall is infiltrated into the soil, up to a point at
which the hydrograph rises and the rainfall begins contributing to the runoff response and the loss becomes a fractional
amount of the rainfall. The parameters of these models are typically calibrated using historical rainfall and streamflow
data (e.g., Brown et al., 2022; Clayton, 2012; Gamage et al., 2015) with either a central tendency value (i.e., mean or
median), or a probabilistic distribution of loss parameters adopted for deterministic design flood estimation approaches
(Rahman et al., 2002; Zhang et al., 2023; Nathan et al., 2003; Gamage et al., 2013; Loveridge and Rahman, 2021;
Ishak and Rahman, 2006).





Under climate change, it has been shown that antecedent soil moisture is changing (Berg et al., 2017; Seneviratne et
al., 2010; Wasko et al., 2021a) and will likely continue to change due to a range of factors. These factors include
increased temperatures, increased rainfall variability and changes in drought duration and frequency (Ukkola et al.,
2020), and changes to the persistence of large-scale ocean-atmospheric mechanisms such as increased persistence of
La Niña (Geng et al., 2023). Any changes in the antecedent soil moisture due to climate change will impact on the
resultant design flood estimate (Ivancic and Shaw, 2015; O'Shea et al., 2021; Quintero et al., 2022).
**4.3.5.2 Systematic review**
While there is ample evidence that climate change will alter antecedent soil moisture conditions, which in turn
modulate flood responses and hence rainfall losses, there have been few studies quantifying how climate change will
affect rainfall loss parameter values. A systematic review found several studies that have assessed the impact of trends
in antecedent moisture conditions and rainfall losses on floods (Earl et al., 2023; Loveridge and Rahman, 2013).
However, we found only two studies projecting rainfall losses, where overall rainfall losses (Ho et al., 2022) and
rainfall loss parameters (Ho et al., 2023, 2022) were projected under climate change. These studies examined the
relationships between total rainfall losses and the parameters of the ILCL rainfall loss model in relation to antecedent
soil moisture in largely unregulated catchments across Australia. Ho et al. (2023) found a consistent negative linear
relationship between the loss parameters and antecedent soil moisture, where increased antecedent soil moisture was
associated with decreased losses. For locations where the relationships between the loss parameters and antecedent
moisture conditions were statistically significant, projections of the loss parameter values were made using projections
of antecedent soil moisture developed by the Australian Bureau of Meteorology (Srikanthan et al., 2022; Wilson et
al., 2022; Vogel et al., 2023). On average, by the end of the century and under RCP 8.5, initial losses were projected
to increase by 5.0 mm (9%) with the interquartile range of the change from 3.3 to 6.3 mm (6%-12%). Continuing
losses were projected to increase on average by 0.45 mm/hr (13%), with an interquartile range of the change of 0.18
to 0.6 mm/hr (8%-23%). To remain consistent with the meta-analysis methodology the above changes, on a per
catchment basis, were standardised using global mean temperature and pooled across Natural Resource Management
Regions (Figure S3, Figure S4). Follow the scaling factors were pooled across RCP to produce the scaling rates shown
in Table 2. Here it was deemed that the variability between regions (refer to Figure 2 from Ho et al. (2023)) was
sufficient to respect regional differences, with events greater or equal to an annual maxima partial duration series
adopted for the development of soil moisture-loss relationships.
**Table 2.** Median scaling factors for loss parameters together presented per degree global temperature change for
clusters of Natural Resource Management Regions (CSIRO and Bureau of Meteorology, 2015), adapted from Ho et
al. (2023). The 'likely' range (corresponding to ~66% range) is presented in parenthesis.

| Natural Resource Management Region | IL (%/°C) | CL (%/°C) |
| --- | --- | --- |
| Southern and South-Western Flatlands | 4.5 (2.0-7.1) | 5.6 (2.5-8.7) |
| Murray Basin | 3.1 (1.0-5.7) | 6.7 (1.5-12.1) |
| Southern Slopes | 3.9 (1.5-7.2) | 8.5 (2.9-15.7) |
| East Coast | 2.0 (0.6-4.3) | 3.8 (1.1-8.0) |
| Central Slopes | 1.1 (0.4-2.2) | 2.0 (-0.5-7.5) |





| | | |
|---|---|---|
| Wet Tropics | 0.8 (-0.4-2.0) | 1.4 (-0.1-4.8) |
| Monsoonal North | 2.4 (1.0-5.4) | 4.4 (3.1-9.5) |


#### 4.3.6 Sea level factors

At the coastal terminus of a catchment, sea levels can modulate flooding, and hence incorporating the appropriate sea level variations in the tail water boundary conditions is an important consideration for coastal and estuarine flood modelling. Moreover, research has shown that extreme rainfall and storm surge processes are statistically dependent, and therefore their interaction needs to be taken into account (Zheng et al., 2013). Despite this, changes in the sea level are not covered in Australia's flood estimation guidance (Bates et al., 2019).

Coastal sea levels vary due to multiple processes that operate on different time and space scales, ranging from astronomical tides and storm surges to long-term sea-level rise due to global warming (McInnes et al., 2016). Astronomical tides occur on a predictable and recurring basis, with relatively consistent frequency. Storm surges, on the other hand, are less frequent and, because they occur in conjunction with severe weather events with low atmospheric pressure, storm surge intensity is related to the strength of the storm. For coastal flooding, the same weather systems that cause storm surges can also produce high rainfall totals and the potential for compound flooding along the coast (Bevacqua et al., 2019; Collins et al., 2019; Zheng et al., 2013).

Both observed and modelled results (Wu et al., 2018; Zheng et al., 2013; Bevacqua et al., 2020) indicate that the dependence between storm surges and extreme rainfalls is strongest in the north and northwest of Australia, followed by the west and northeast of Australia. It is weak and/or statistically not significant on the northeastern tip of Queensland, along the southeast coast of Western Australia, along small parts of the South Australian coastline, and along the eastern part of the Victorian coast near Bass Strait. As the co-occurrence of extreme rainfall with extreme storm surge is similar to the co-occurrence of runoff with storm surge (Bevacqua et al., 2020), methods for incorporating this dependence are in included in Australia's flood guidance (Westra et al., 2019) – despite sea level rise not being included. In the northern part of the continent, coincident extremes are most likely due to the occurrence of tropical cyclones. Along the southwest and southern coastline, coincident extremes are most likely due to extratropical lows and associated cold frontal systems during the winter half year. Along the southeast coast, coincident events are most likely due to cut-off lows or frontal systems (Wu et al., 2018).

While coincident flood studies have focussed on the coincidence of rainfall or runoff events with storm surges or storm tides, other factors can also affect regional sea level variability on differing time scales. For example, coastally-trapped waves (CTWs) can cause sea level variability along Australia's extratropical coastline on timescales from weeks to months, with amplitudes correlating with continental shelf width and ranging from 0.7 m along the south coast to 0.05–0.10 m along the east coast (Eliot and Pattiaratchi, 2010; Woodham et al., 2013). In some locations, seasonal-scale sea level variations are an important consideration. For example, the Gulf of Carpentaria experiences a significant annual sea level range of about 0.8 m, which is driven mainly by the seasonal reversal of the prevailing winds. On interannual time scales the El Niño-Southern Oscillation causes sea level variations with higher (lower)





than average sea levels during La Niña's (El Niño's), which have a maximum range in the Gulf of Carpentaria and
decrease in magnitude with distance anticlockwise around the coastline (White et al., 2014; McInnes et al., 2016).
Sea-level rise (SLR) is increasing the frequency of coastal flooding (Hague et al., 2023). Over the period from 2007
to 2018 sea levels rose at an average rate of $3.6 \pm 1.7$ mm/yr based on a global network of tide gauge records, and
$3.8 \pm 0.3$ mm/yr based on satellite altimeters (Wang et al., 2021). Over the period 1993-2018 in the same two datasets,
the rates of SLR were $0.063 \pm 0.120$ and $0.053 \pm 0.026$ mm/yr$^2$, respectively, indicating that SLR is accelerating
(Wang et al., 2021). In Australia, the rate of SLR based on Australian gauges from the ANCHORS dataset, with at
least 50 years of data over 1966 to 2019, was 1.94 mm/yr, and over 1993 to 2019 was 3.74 mm/yr (Hague et al., 2022).
With the increase in the flood frequency over the observational record, mainly because SLR is increasing the height
of the tides with ongoing SLR, flooding events will become increasingly predictable (Hague et al., 2023).
**Table 3.** Sea-level rise (m) relative to 1995-2014 for CMIP6 and associated 5-95% confidence intervals (Source:
Table 9.9 in Fox-Kemper et al. (2021)).

| Scenario | 2050 | 2100 | 2150 |
|---|---|---|---|
| SSP1-1.9 | 0.18 (0.15-0.23) | 0.38 (0.28–0.55) | 0.57 (0.37–0.86) |
| SSP1-2.6 | 0.19 (0.16-0.25) | 0.44 (0.32–0.62) | 0.68 (0.46–0.99) |
| SSP2-4.5 | 0.20 (0.17-0.26) | 0.56 (0.44–0.76) | 0.92 (0.66–1.33) |
| SSP3-7.0 | 0.22 (0.18-0.27) | 0.68 (0.55–0.90) | 0.92 (0.66–1.33) |
| SSP5-8.5 | 0.23 (0.20-0.29) | 0.77 (0.63–1.01) | 1.98 (0.98–4.82) |
| SSP5-8.5* | 0.24 (0.20-0.40) | 0.88 (0.63–1.60) | 1.98 (0.98–4.82) |

*includes additional 'low confidence' processes
Projections of future SLR provided by the IPCC in its Sixth Assessment (AR6) report for a set of future greenhouse
gas emission pathways termed SSPs (Fox-Kemper et al., 2021) are summarised for the years 2100 and 2150 in Table
3, along with their associated uncertainties. Note this only refers to mean sea level changes; processes associated with
extreme sea levels such as storm surge and wave set-up that may be used in design flood estimation are not included.
The processes included in the projections are assessed by the IPCC to be of '*medium confidence*' and include changes
due to thermal expansion, the mass balance of glaciers and ice sheets, and terrestrial water storage. The IPCC also
provide scenarios they assess to have '*low confidence*' of occurring on the time scales considered, such as dynamical
processes that could lead to more rapid disintegration of the ice sheets (DeConto and Pollard, 2016; Fox-Kemper et
al., 2021).
Changes to weather and circulation patterns will also potentially change storm surge and wave patterns, altering
compound flooding. For example, Colberg et al. (2019) investigated changes in extreme sea levels around Australia
by forcing a hydrodynamic model with winds and surface pressure from four GCMs run with an RCP 8.5 emission
scenario over the periods 1981-1991 and 2081–2099. The largest positive extreme sea-level changes were found over
the Gulf of Carpentaria due to changes in the northwest monsoon, while mainly negative changes in seasonal
maximum sea levels up to -5.0 cm were found along Australia's southern coastline due to the projected southward
movement of the subtropical ridge and associated cold frontal systems, with these results broadly consistent with other
studies (Colberg and McInnes, 2012; Vousdoukas et al., 2018). Extreme coastal sea levels are also affected by wave



breaking processes that cause wave setup (O'Grady et al., 2019), with the 1 in 100 AEP wave height projected to
increase by 5 to 15% over the Southern Ocean by the end of the 21[st] century (2081-2100), compared to the 1979–2005
period (Meucci et al., 2020). Finally, coastal erosion of sandy shorelines and estuaries under SLR will also contribute
to changes in coastal flooding patterns. Historical coastline movement around the Australian coast has been evaluated
through analysis of satellite images using a technique to filter satellite pixels to a consistent tide datum (Bishop-Taylor
et al., 2019, 2021). Over 22% of Australia's non-rocky coastline shows trends of both significant coastal retreat or
growth since 1988, with most change (15.8%) occurring at rates greater than 0.5 m/yr.

### 5. Discussion

From this systematic review on climate change science relevant to design flood estimation in Australia, it emerged
that most published research relates to changes in extreme rainfall intensity, and hence the IFDs and PMPs that are
used in event-based modelling. Here we aim to resolve the understanding of changes in extreme rainfall with
methodologies applied for design flood estimation. Following this, factors that were beyond the scope of this review
are acknowledged and a summary of future research priorities are presented.

### 5.1 Aligning evidence of changes in extreme rainfall with design flood estimation

Although we were unable to quantify the increase in extreme rainfall across a range of frequencies, studies  using
rainfall-temperature scaling (Wasko and Sharma, 2017b), historical trends (Wasko and Nathan, 2019; Jayaweera et
al., 2023), and climate change projections (Pendergrass and Hartmann, 2014; Pendergrass, 2018; Carey-Smith et al.,
2018), all show that the rate of rainfall increase with increasing rarity. Operational methods employed to estimate
PMPs are restricted to the consideration of thermodynamic increases in the moisture holding capacity through changes
in the moisture adjustment factor (Visser et al., 2022). However, short duration extremes (sub-daily) have been shown
to increase at rates greater than CC scaling both for Australia (presented herein) and globally (Fowler et al., 2021).
There is no obvious physical explanation for why changes to sub-daily PMP estimates should differ from other studies
on sub-daily extreme precipitation. Synthesising the evidence, it appears that (1) increases in long duration extreme
rainfalls should plateau to a rate of increase commensurate with the PMP, which is likely to be increasing at the CC
rate for daily rainfall; and (2) increases in short duration rainfall, in the absence of research into changes in PMP for
sub-daily durations, should increase at the rate of the short duration rainfall extremes. It is plausible that PMPs will
increase in line with short duration rainfall extremes due to an increase in storm efficiency, which is a well-established
mechanism in short duration rainfall due to latent heat release increasing buoyancy (Lenderink et al., 2019). Further
increases above those simply owing to thermodynamics are also possible due to reductions in the speed of lateral
storm movement.
It is clear that increases in the order of 2-3 times the CC rate are a possibility for design rainfalls throughout Australia,
with greater potential increases in the north than in the south. This is generally related to the occurrence of convective
storms, such as severe thunderstorms that can cause short duration (e.g., less than about 6 hours) localised extreme
rainfall. Although current Australian climate modelling studies are generally not able to simulate the processes
relevant to these convective rainfall extremes, as they are not run at convection-permitting scales, the observation-
based increases are broadly consistent with theoretical expectations based on increased rainfall efficiency from



increased condensation for enhanced convection. Changes greater than the CC rate due to more efficient convective
processes can also be relevant for annual maxima longer than that of typical thunderstorms. For example, the highest
recorded daily rainfall at Adelaide occurred over a period of only two hours due to a thunderstorm (Ashcroft et al.
2019). This means that increases greater than the CC rate may also be plausible for more widespread and longer
duration rainfall extremes, such as multiday-duration events associated with TCs in near-coastal northern regions and
ECLs in eastern and south-eastern regions that sometimes contain deep moist convection (Callaghan and Power,

948   2014).

### 5.2 Systematic review and meta-analysis considerations

We have attempted to minimise biases where possible. Consistent with the IPCC methodologies, a multiple-lines-of-
evidence approach was adopted  considering historical changes, future projections, and physical argumentation. As
such, inherent methodological biases, such as issues associated with hypothesis testing favouring the null hypothesis,
would only apply to a proportion of the evidence. Next, analyses to inform assessment reports such as the IPCC and
CCIA often present projections separately from any claims of significance and are not required to demonstrate
originality of contribution; therefore, these studies are less likely to be affected by both the hypothesis testing and
publication biases - noting that hypothesis testing bias and publication bias would be expected to act in opposing ways.
Finally, researcher biases were addressed by having two researchers independently evaluate each reference for their
area, and by adopting a systematic review framework so that publications are not just chosen on the basis of a
researcher's prior knowledge or expectation. This was also addressed in the meta-analysis by sensitivity testing the
results through multiple researchers independently weighting evidence. The outcomes of the per-researcher analyses
were consistently similar.
In addition to the review biases, the limitations of each line of empirical evidence need to be acknowledged. It can be
difficult to identify a climate change signal in observational records, firstly due to the small signal to noise ratio, but
secondly due to the difficulty of obtaining high quality instrumental data (Hall et al., 2014). For example, it is difficult
to detect a statistically significant change resulting from Clausius-Clapeyron scaling at a single rain gauge based on
observed warming rates and typical record lengths (Westra et al., 2013), such that the absence of a statistically
significant result does not necessarily imply the absence of a trend. Single site studies were hence given low weighting
in the meta-analysis. Further, it needs to be acknowledged that a historical trend can only be extrapolated to the future
by assuming the causal relationship remains unchanged, which may not be true (Wasko, 2022; Zhang et al., 2022).
The second line of evidence was the empirical relationship between day-to-day variability in rainfall and surface air
or dew-point temperature for high quantiles of the distribution. Although robust relationships have now been
established globally (Ali et al., 2018, 2021a, b), debate remains over the use of these day-to-day scaling relationships
for projection as near-surface conditions may not reflect key factors in rainfall production, such as potential future
changes in the vertical temperature profile of the atmosphere or changes to rainfall efficiency. The limitations of the
above two sources of evidence can be somewhat overcome by the third line of evidence, that is, climate modelling
which explicitly models atmospheric conditions; however, it needs to be acknowledged that not all processes related
to rainfall are resolved (François et al., 2019). Global as well as many regional climate models have large spatial scales



compared to some of the physical processes associated with rainfall (e.g., localised convection) and struggle to
represent some aspects of rainfall occurrence (e.g., short-duration convective rainfall extremes, such as produced by
thunderstorms). Hence, recommendations here are based on an expert evaluation that has combined all the key lines
of evidence, recognising the known limitations of any single line of evidence.

### 5.3 Factors omitted and recommendations for future work

This review focussed on a set of salient factors relevant to design flood estimation, and hence there are some aspects
that are not covered. Australia has three small regions located in the south-east of the country that currently sustain
snowpacks over the winter period: the Snowy Mountains region in southern New South Wales, the Victorian Alps,
and the Tasmania highlands. Studies of the contribution of rain-on-snow events to flood risks have been undertaken
using simple water budget approaches (Stephens et al., 2016; Nathan and Bowles, 1997). While rain-on-snow events
dominated the generation of more frequent floods (≥ 1 in 50 AEP), they were less important for more extreme events.
The key engineering design focus in these regions is related to the overtopping risks of hydroelectric dams; and as
such, snowmelt floods are considered a localised issue for Australia and are not covered in the national flood guidelines
(Ball et al., 2019a).
Design flood practice in Australia, as elsewhere in the world, generally adopts areal lumped temporal patterns in
combination with a fixed spatial pattern. The information available to characterise this variability is very limited and
this dearth of evidence poses problems for design flood estimation under stationarity assumptions and limits our ability
to estimate the impacts of climate change on flood risks. With climate change, it is important to correctly reflect
changes in spatial and temporal correlation structures and transition probabilities, particularly for large catchments,
which are sensitive to spatial variability in rainfalls, or for such applications as the design of linear infrastructure such
as railways and major highways (Le et al., 2019). It can be expected that the only way the impacts of climate change
can be considered on the spatio-temporal patterns of extreme rainfall is through a combination of physical modelling
(e.g., Chang et al. 2016) and careful regional pooling (e.g., Visser et al. 2023). Finally, it is also worth noting that no
attention is given to the impact of climate change on factors exogenous to storm climatic drivers. An example of this
is the assessment of water levels in dams, or surcharge flooding from sewer networks. Climate change impacts are the
result of a complex mix of water demands and water management strategies (not to mention longer-term climatic
conditions) that are not a function of storm events, but such analyses require tailored approaches for which it is difficult
to provide general guidance.
While there remains a need for guidance on how to perform flood frequency analysis and continuous simulation under
climate change, a lack of consensus remains on how best to perform these, a point noted by previous authors (Schlef
et al., 2023). Although recent research has shown that bias-correcting for changes to long-term persistence (interannual
variability) is critical for hydrological studies (Vogel et al., 2023), a standard approach for deriving time series rainfalls
under climate change remains a research priority (Dale, 2021). While event-based methods allow the adjustment for
climate change of the primary flood drivers, it remains a research gap to understand under climate change to which
drivers the design flood estimate is most sensitive to – a problem that may lend itself to being addressed by
sensitivity/stress testing (Hannaford et al., 2023) or applying a storyline approach in flood estimation (de Bruijn et al.,





2016; Shepherd et al., 2018; Hazeleger et al., 2015) but this requires an understanding of the causal mechanisms of flood events which remains limited in Australia (Wasko and Guo, 2022).

Finally, the development of climate models with the ability to resolve convection processes in other parts of the world (Chan et al., 2020, 2016) suggests the potential for improved simulations and projections of short duration rainfall extremes in Australia. Improved projections of short duration extreme rainfalls would be particularly valuable given the understanding that these events are increasing at a greater rate than long duration rainfalls. However, a substantial constraint to modelling convection processes are the computationally intensive modelling efforts required to cover the geographic expanse of Australia.

## 6. Summary and conclusions

This paper describes a review of the scientific literature as it relates to the impact of climate change on design flood estimation for Australia. To ensure the review is reproducible and to minimise the potential for bias, we adopted the framework of a systematic review. To be included, studies needed to pertain to either flood risk drivers or a measure of the flood hazard itself; how these are impacted on by climate change; and be relevant to Australia. As design flood estimation is undertaken using similar methods across the world, knowledge from relevant international research was included in addition to the systematic review, particularly in instances where local evidence was limited. The conclusions of this systematic review, as they relate to the methods for design flood estimation, are described below and summarised in Table 4:

1.  There is a general absence of a scientifically defensible methodology for performing flood frequency analysis in the context of projections for a future climate. The projection of a historical temporal trend is not recommended, with many studies arguing that any non-stationary flood frequency analysis should ensure that the statistical model structure is representative of the processes controlling flooding. But as flood processes change with climate change, and with historical data likely to be influenced by other drivers such as land-use change, extrapolating historical trends into the future is not considered a viable method for developing future estimates of flood risk.

2.  The use of continuous simulation for flood frequency projections requires downscaling and bias-correction of GCM outputs to derive hydrologic inputs such as rainfall that represent a future climate. Due to the complexity in extracting GCM data and appropriately transforming the GCM data to the local scale, approaches of projecting flood frequency through continuous simulation are likely to, at least in the near term, remain limited to research applications. Dale (2021) notes that a standard approach for deriving time series rainfalls under climate change remains a research priority. If continuous simulation is to be applied, careful attention needs to be paid to ensuring downscaling and bias-correction methodologies accurately correct both extreme rainfall and long-term variability (persistence) characteristics that are important to hydrological applications (Vogel et al., 2023).

3.  The primary input into event-based modelling is the IFD rainfall. The IPCC states that the frequency and intensity of heavy precipitation events have likely increased due to climate change (Seneviratne et al., 2023).





Here we find that both daily and sub-daily rainfall are increasing with warming, with the rate of increase
greater for shorter durations. Moreover, there is emerging evidence that the rarer the rainfall, the greater
increase, and that increases in sub-daily rainfall extremes are greater in the tropics. However, there is
currently not enough quantitative evidence across different exceedance probabilities or geographic zones to
quantify projections of extreme rainfall across different regions of Australia.
4.    Both literature from Australia and across the world provides a consensus view that the PMP is likely
increasing at the CC rate for daily rainfall. Despite no research on changes in the PMP at the sub-daily scale,
it appears extreme rainfall increases plateau with increasing severity (Pendergrass, 2018). Hence, as storms
intensify with climate change due to latent heat release, it can be assumed that changes above the CC scaling
rate for the rarest of extreme rainfalls at the sub-daily scale can be a taken as representative of changes to the
PMP for similar durations.
5.    Evidence exists to suggest that temporal patterns will become more front loaded and intense with climate
change, but evidence for changes in spatial patterns is not conclusive, with changes likely to vary with
weather system. Currently, there is no adopted methodology for how to incorporate these changes into design
flood estimation, or assessment of the impact incorporating such changes will have on the design flood
estimate.
6.    With climate change, across Australia, catchment soil moisture conditions prior to an extreme rainfall event
are largely becoming drier and hence losses are projected to increase (Ho et al., 2023). These changes in
antecedent moisture conditions have been shown to modulate both historical and future frequent floods, with
the impact on rarer floods diminished (Wasko and Nathan, 2019; Wasko et al., 2023).
7.    Sea levels have risen across Australia, impacting estuarine flooding, and resulting in much of Australia's
coastline retreating. With future increases in sea level projected with global warming, estuarine flooding
events will become increasingly predictable. However, the changes to the interaction between coastal sea
levels and pluvial riverine flooding remain poorly understood.

1074       **Table 4.** Conclusions of systematic review of climate change science relevant to Australian design flood
1075                                       estimation.

| Method | Quantity | Findings |
|---|---|---|
| Flood frequency analysis | Streamflow | No defensible methods were identified for factoring in climate change into flood frequency estimates. |
| Continuous simulation | Rainfall and evaporation | At present, there are limited studies that describe how to generate realistic time series of weather suitable for flood risk estimation. Further research is required before there is a continuous simulation method suitable for standard practice in design flood estimation. |
| Event-based estimation | Extreme rainfall (up to and including the PMP) | Heavy precipitation events have increased and will continue to increase due to climate change, with the highest rates of increase associated with short-duration rainfall. Australia-wide estimates (including a central estimate and 'likely' range) are provided in Table 1, varying by duration. Whilst there is reason to believe that scaling rates will vary both geographically (with higher rates in the north of Australia) and by exceedance probability (with higher rates for rarer |





| | events), insufficient evidence was available to quantify the differences in projected changes with location and AEP. It is, however, likely that these changes are within the uncertainty intervals provided in Table 1. |
|---|---|
| Temporal patterns | Temporal patterns may become more front-loaded, with increases in peak intensities with climate change, but research on the impact of these changes on design flood estimation is lacking. |
| Areal reduction factors | Evidence for changes in spatial patterns with climate change is not conclusive. |
| Antecedent conditions | For Australia there is evidence of drying antecedent conditions, meaning increased losses in design flood estimation, but this research has not yet been translated for use in design flood estimation. |
| Sea level interaction | Whilst there is significant evidence that sea levels are increasing and will continue to increase due to climate change, the changes to the interaction between high ocean levels (due to the combination of high astronomic tides and storm surges) and heavy rainfall events remains poorly understood. |


To synthesise findings for changes in rainfall intensity quantitatively, a meta-analysis was performed. The uncertainty
presented in the meta-analysis serves to demonstrate that a single line of evidence is not sufficient for deciding on the
impact of climate change. As studies vary widely in the approaches and assumptions, multiple lines of evidence should
be considered in decision making related to climate change, and the latest climate science reviewed in decision making.
Although Australia is not a climatically homogenous nation, there does not exist enough information to distinguish
extreme rainfall changes regionally, highlighting the need for continental-scale, high-resolution (convection-
permitting) modelling efforts to help understand the impact of climate change on extreme rainfalls. Nevertheless, there
is now a large body of work on changes to flood drivers as a result of climate change, and whilst significant uncertainty
remains, this work can be used to form the basis for producing improved methods for defensible estimates of future
flood risk.
**Code availability**
Code used to calculate warming levels can be found at https://github.com/traupach/warming_levels.
**Author contribution**
**CW** Conceptualization, Writing – original draft preparation. **SW** Conceptualization, Methodology, Writing –
original draft preparation, Writing – review & editing. **RN** Conceptualization, Writing – original draft preparation.
**AP** Writing – original draft preparation, Formal analysis. **TR** Writing – original draft preparation, Formal analysis.
**AD** Writing – original draft preparation. **FJ** Writing – original draft preparation. **MH** Writing – original draft
preparation. **KLM** Writing – original draft preparation. **DJ** Writing – review & editing. **JE** Writing – review &
editing. **HJF** Writing – review & editing. **GV** Writing – review & editing.
**Competing interests**
The authors declare that they have no conflict of interest.
**Acknowledgments**
This work was supported by Department of Climate Change, Energy, the Environment and Water. Conrad Wasko
acknowledges support from the Australian Research Council (DE210100479). Acacia Pepler, Andrew Dowdy, Jason



Evans, and Timothy Raupach acknowledge funding from the Climate System Hub of the Australian National
Environmental Science Program. Fiona Johnson is supported by the ARC Training Center in Data Analytics for
Resources and Environments (IC190100031).

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
