# Peer review of "A systematic review of climate change science relevant to"

_Hydrology and Earth System Sciences, 2023_

## Author Comment (AC1)

**Reply to RC1 (reviewer comments in grey)**

In this work, the authors revised the state of the art on the impact of climate change on flood risk, posing major attention to Australia.

The review is very well written and in line with the scope of the Journal, and there is no need for significant changes before publication.

In my opinion, the manuscript can benefit from a few minor changes:

We thank the reviewer for their assessment and thoughtful comments. Please find below our reply in blue and proposed modifications underlined.

- in light of having a coherent structure, maybe it is worth adding the "Systematic review" subsection to sections 4.3.1 and 4.3.6

Sections 4.3.1 and 4.3.6 were excluded from the strict requirements of the systematic review as, although relevant to design flood estimation, they are not explicitly included in Australia's flood guidance. We only commented on sea level rise section at Line 161. To clarify, Line 161 would be expanded to explain why Section 4.3.1 (background to the processes affecting Australian extreme rainfall) is excluded. Also, the beginning of Sections 4.3.1 and 4.3.6 would have a sentence added at their start to explain why they are not part of the systematic review, but still included in section 4.3.

- please add a reference to the sentence at lines 557-559, to justify the "…it is well known…" part

Agreed. We would remove "it is well known" and also add a reference to the following: Linacre, E., Geerts, B., 1997. Climates & Weather Explained. Routledge, London; New York.

- please consider adding a (graphical) summary of the review in terms of total number of articles per year, per topic, etc. I know that those numbers are provided within the manuscript, but maybe having a figure summarizing them can help readers

Agreed. We would provide a stacked bar chart summarising the literature in Table S2. The x-axis would be the year, and the bars would be coloured by the topic of the manuscript.

- I would like to see some more comments on the reproducibility/transferability of this study. As the results are very much connected with the authors' expertise, what are the key problems in reproducing (or eventually updating) this review? Are the used methods transferable to other countries? Do you think that the results might be influenced by the experts involved?

Agreed. Text would be added to the discussion, likely in an additional section, discussing the barriers to adoption and transferability, also placing this study in the context of global attempts to update flood guidance.

- In section 5.2 you briefly addressed the biases involved in the study, and in lines 960-961 you said "The outcomes of the per-researcher analyses were consistently similar". Is it possible to have a more quantitative picture of such a similarity, as well as some more details on the sensitivity testing (line 959)?

Agreed. A table will be added to the supplementary material of the individual researcher quantitative results and a further summary of the methods of averaging adopted by each researcher would also be provided.

I am confident that addressing the above points could help in further improving an already very good manuscript.

Thank you again for this assessment and the time taken to thoughtfully provide feedback on our manuscript.

---

## Author Comment (AC2)

**Reply to RC2 (reviewer comments in grey)**

**Review of Wasko et al, "A systematic review of climate change science relevant to Australian design flood estimation"**

**Overall assessment**

This is an exceptionally thorough and well-written review, which I think will be of interest to a wide readership. The authors methodically review the different approaches for design flood estimation, with a specific focus on the Australian context. They apply objective methods to assess the level of consensus of different sources of evidence. The findings are mostly uncontroversial, and some are quite important, such as the fact that no defensible method currently exists for factoring climate change into flood frequency estimates. The summary of changes in rainfall with warming is particularly interesting, as is the table summarising all the findings (Table 4).

We thank the reviewer for their assessment and thoughtful comments. Please find below our reply in blue and proposed modifications in underlined.

**Main comments**

L.81. The authors state that there has been "little research undertaken on the non-stationarity of other factors affecting the design flood estimate". There has been considerable research on a range of nonstationary factors affecting large floods, so this statement seems a little surprising. Does it matter that the research has focussed on nonstationary factors affecting general floods rather than design floods? Presumably the challenge is more about understanding how to incorporate the nonstationarity in the design estimates, rather than a lack of research on nonstationary factors.

The reviewer is correct. Our intention was to state that most research does not explicitly focus on the impacts of climate change on the inputs into design flood estimation. This sentence would be reworded to explicitly state that although much research has been undertaken into the factors affecting flooding, they are often not directly comparable or translated to the inputs used in design flood estimation using event-based models (the exception to this being peak rainfalls).

L.97-145. Section 2 is a very helpful overview of the main methods, but it provides almost no citations. I think it would be helpful to include at least the key references for each of the methods described. It might also be helpful for the reader to provide a summary table of the strengths/limitations of each method (I leave this decision with the authors).
The paper reviews different methods for design flood estimation based on streamflow and based on rainfall. At various points in the review, the authors seem to suggest that the streamflow-based flood projections are either too complex (L.1040) or too uncertain (L79) for practitioners to apply. I wonder if it might be worth reflecting a little more on the adequacy of different approaches, and the limitations of the simpler precipitation-based approaches (which make up a very large part of the review). They may work well for certain types of flooding (e.g. pluvial floods in an urban context), but is it fair to say they may not work so well for fluvial flooding, where other factors such as changes in groundwater storage need to be taken into account?

In response to the reviewer comments, we would:

(1) Add key references to each of the methods (i.e., continuous simulation, flood frequency analysis, and event-based simulation).

    (2) Lines 142-145 would be expanded to discuss the strength/limitations of each method (this would tie in with Section 5.3 where the limitations with respect to climate change are briefly discussed).

    (3) To the above point and the reviewer's suggestion, Line 1006-1015 would be expanded to explicitly discuss the adequacy but also limitations of each method in the context of climate change. Particularly at Line 1007 where we state there is a lack of consensus on the best method to adopt under climate change, but do not explain why this is the case.

Pg.17-18. It is clear that a lot of effort went into producing the 'best estimates of central scaling rates' in which the authors independently assigned weights to different studies to arrive at a median estimate and 66% range. The Supplementary table is an impressive exercise in collating and trying to synthesize information from 179 different sources. However, it is a little difficult to see how the summary values in Table 1 were arrived at, since the weights are not provided and the thresholds shown in Table 1 (<1hr or >24hr) don't match those shown in Figure 2 (<6hr, >12hr) (perhaps I missed why Table 1 and Figure 2 differ). It would be helpful to provide more data on how the values were obtained, so it doesn't seem too subjective.

Agreed. A table will be added to the supplementary material of the individual researcher quantitative results and a further summary of the methods of averaging adopted by each researcher would also be provided. In addition, the labelling on Figure 2 will be updated to match Table 1.

**Minor comments**

L.99 "the primary differences between methods relating to where in the causal chain of flooding the data are obtained, and where the probability model is fitted" could be rephrased for greater clarity, perhaps being more specific.

Agreed. This sentence would be removed and replaced with a brief introduction to the design flood estimation methods to help introduce the text that follows.

L.139. The term "efficacy" in the caption of Figure 1 is a little ambiguous and could be clarified.

Agreed. Efficacy would be replaced with words that emphasize that in the highlighted AEP ranges the methods show the most utility.

L.172/183 "average effect size" – please specify the effect size of what.

This will be replaced with "magnitude of extreme rainfall depth change" or words to that effect.

L.177 "with variation to storm duration .. and location preserved" could be rephrased for clarity.

Agreed. These factors were considered as additional variables, and the text will be amended to make this clear.

L.186 "was weighted by"?

This typo will be fixed as per the reviewer's suggestion.

L.206-211. Recent work has shown that groundwater is a more important driver of flooding than either antecedent soil moisture or antecedent extreme rainfall (see work by W.Berghuijs); this may be worth mentioning.

The text will be amended to add the reviewer's suggestion.

L.237-240 "even the use of physically-based covariates is problematic as the covariates should capture the differing processes": please clarify/elaborate. Also, which of the "statistical associations may not remain constant with climate change"?- it is worth being more explicit.

Here we wished to state that although the covariate may be physically linked to the processes of climate change, if the covariates does not explicitly model the physical relationship driving the change, then extrapolation may result in incorrect estimates. And even if this is physical relationship is captured, there is still no guarantee that the governing processes will be the same in the future. The wording will be rephrased to make this clear.

L.432 "their application to the future period remains untested": the phrasing is a little odd; application to the future is always untested. Do you mean their predictive ability (to predict out of sample) is untested?

This is correct. The suggested rephrasing will be adopted.

L.957-961. I am not sure how helpful it is to tell the reader that the papers were assessed independently and through weighting of evidence, if the outcomes of those analyses are not presented. It's a bit like asking the reader to simply *trust* that the analysis is objective.

Agreed. As per the response above, a table will be added to the supplementary material of the individual researcher quantitative results and a further summary of the methods of averaging adopted by each researcher would also be provided. In addition, the labelling on Figure 2 will be updated to match Table 1. This section would then reference the additional results table.

L.1068 "the impact on rare floods diminished" could be rephrased for clarity.

This will be rephrased to state that the impact on rarer floods is lesser.

**Figures**

Figure 1. Nice summary figure. The labels of the x-axis could explained in the caption. The "S" shape of the curve could also be explained.

Figure 2. Again, very interesting (and novel) summary figure. I don't think 2xCC and 3xCC are defined.

The omissions referred to by the reviewer will be added.

---

## Author Response (AR1)

Dear editor,

We thank you for the opportunity to revise our manuscript. Please find our responses to the reviewer and community comments below. Our responses are shown in brown with line numbers referring to the 'clean' version of the manuscript.

We have also proofread our document thoroughly and fixed all typos and improved wording for clarity. Please refer to the track changes document for this minor changes.

**RC1**: 'Comment on hess-2023-232'

In this work, the authors revised the state of the art on the impact of climate change on flood risk, posing major attention to Australia.

The review is very well written and in line with the scope of the Journal, and there is no need for significant changes before publication.

In my opinion, the manuscript can benefit from a few minor changes:

We thank the reviewer for their assessment and thoughtful comments. Please find our responses in brown below.

- in light of having a coherent structure, maybe it is worth adding the "Systematic review" subsection to sections 4.3.1 and 4.3.6

Sections 4.3.1 and 4.3.6 were excluded from the strict requirements of the systematic review as, although relevant to design flood estimation, they are not explicitly included in Australia's flood guidance. This has been clarified in Lines 176-180 which now reads:

"We note that the impact of factors related to sea level (Section 4.3.6), although included in the review, was excluded from the requirements of the systematic review as it is not explicitly part of Australia's flood guidance as it relates to climate change (Bates et al., 2019). Similarly, the introductory section on the processes affecting changes in extreme rainfall in Australia (Section 4.3.1) was excluded from the stricter systematic review requirements."

Further, Sections 4.3.1 and 4.3.6 now begin with explicit statements that these sections were not part of the systematic review (see Lines 332, 865).

- please add a reference to the sentence at lines 557-559, to justify the "…it is well known…" part

We have removed "it is well known" and added a reference to the following: Linacre, E., Geerts, B., 1997. Climates & Weather Explained. Routledge, London; New York (now Line 572).

- please consider adding a (graphical) summary of the review in terms of total number of articles per year, per topic, etc. I know that those numbers are provided within the manuscript, but maybe having a figure summarizing them can help readers

Figure 2 in the updated manuscript now presents a summary as per the reviewer's suggestion.

- I would like to see some more comments on the reproducibility/transferability of this study. As the results are very much connected with the authors' expertise, what are the key problems in

reproducing (or eventually updating) this review? Are the used methods transferable to other countries? Do you think that the results might be influenced by the experts involved?

As per the authors suggestion, a paragraph has been added at Lines 1000-1015 discussing the barriers to adoption and transferability, as well as placing the work here in the context of other efforts.

- In section 5.2 you briefly addressed the biases involved in the study, and in lines 960-961 you said "The outcomes of the per-researcher analyses were consistently similar". Is it possible to have a more quantitative picture of such a similarity, as well as some more details on the sensitivity testing (line 959)?

As per the reviewer's suggestion Table S3 now contains the quantitative results of each researcher and their methodology.

I am confident that addressing the above points could help in further improving an already very good manuscript.

Thank you again for this assessment and the time taken to thoughtfully provide feedback on our manuscript.

**CC1**: 'Comment on hess-2023-232', Rasmus Benestad, 08 Nov 2023

Thanks for this review. In my group, we have tried new approaches for deriving information about heavy precipitation with an emphasis on creating robust, albeit approximate, results. This is motivated by the fact that the models are never perfect and have a minimum skillful scale. Also, downscaling is associated with uncertainties connected to assumptions and caveats (IPCC AR6). I would suggest including a more explicit account for why global warming is expected to affect rainfall patterns in the introduction. A common explanation for more extreme rainfall is higher temperatures and increased evaporation and moisture-holding capacity of the air (thermodynamics). This may not be the only factor, as a reduction in the global surface receiving 24-hr precipitation (a dynamic factor) also may explain more intensive rainfall (e.g. DOI:10.1371/journal.pclm.0000029 and DOI:10.21203/rs.3.rs-3198800/v1).

One approach is to try to downscale the shape of the curves describing pdfs, probabilities or IDFs, and two key parameters seem to be the wet-day mean precipitation and the wet-day frequency (e.g. DOI:10.1088/1748-9326/ab2bb2 for describing probabilities). Such an approach is limited to "moderate extremes", but on the other hand, there are few data points far out in the tails and estimation of extreme statistics are notoriously messy. The same parameters can also be used as a rule.of.thumb approximation of IDFs (e.g. DOI:10.1088/1748-9326/abd4ab), and even if one is not satisfied with their skill, they may nevertheless serve as a benchmark. Another approach towards predicting the shape of IDFs is through PCA (DOI:10.5194/hess-27-3719-2023).

If the authors want to carry out a systematic review of climate change science relevant to Australian design flood estimation, then it would stand stronger by including lateral ideas presented in the said studies. They represent an attempt to do something similar in Norway and other parts of the world.

We thank the reviewer for their thoughtful comments. We have now:

- Added more scientific background in the introduction (refer to Lines 58-69) to the increases in extreme rainfall.
- Added a paragraph at Lines 1000-1015 discussing the barriers to adoption and transferability, as well as placing the work here in the context of other efforts.

**Review of Wasko et al, "A systematic review of climate change science relevant to Australian design flood estimation"**

**Overall assessment**

This is an exceptionally thorough and well-written review, which I think will be of interest to a wide readership. The authors methodically review the different approaches for design flood estimation, with a specific focus on the Australian context. They apply objective methods to assess the level of consensus of different sources of evidence. The findings are mostly uncontroversial, and some are quite important, such as the fact that no defensible method currently exists for factoring climate change into flood frequency estimates. The summary of changes in rainfall with warming is particularly interesting, as is the table summarising all the findings (Table 4).

We thank the reviewer for their assessment and thoughtful comments. Please find our responses in brown below.

**Main comments**

L.81. The authors state that there has been "little research undertaken on the non-stationarity of other factors affecting the design flood estimate". There has been considerable research on a range of nonstationary factors affecting large floods, so this statement seems a little surprising. Does it matter that the research has focussed on nonstationary factors affecting general floods rather than design floods? Presumably the challenge is more about understanding how to incorporate the nonstationarity in the design estimates, rather than a lack of research on nonstationary factors.

The reviewer is correct. Our intention was to state that most research does not explicitly focus on the impacts of climate change on the inputs into design flood estimation. This sentence now states (Line 93): "Further, while much research has been undertaken on understanding the non-stationarity of flooding, the research is not often directly comparable or translatable to the approaches and methods used in design flood estimation, for example in the case of temporal and spatial patterns of rainfall or the influence of antecedent conditions on rainfall losses (Quintero et al., 2022)."

L.97-145. Section 2 is a very helpful overview of the main methods, but it provides almost no citations. I think it would be helpful to include at least the key references for each of the methods described. It might also be helpful for the reader to provide a summary table of the strengths/limitations of each method (I leave this decision with the authors).
The paper reviews different methods for design flood estimation based on streamflow and based on rainfall. At various points in the review, the authors seem to suggest that the streamflow-based flood projections are either too complex (L.1040) or too uncertain (L79) for practitioners to apply. I wonder if it might be worth reflecting a little more on the adequacy of different approaches, and the limitations of the simpler precipitation-based approaches (which make up a very large part of the review). They may work well for certain types of flooding (e.g. pluvial floods in an urban context), but is it fair to say they may not work so well for fluvial flooding, where other factors such as changes in groundwater storage need to be taken into account?

We acknowledge and agree with the reviewer's primary concern that our wording suggested inadequacy of streamflow-based procedures without adequate justification or contextualisation. We have made the following edits to our manuscript:

(1)  Added key references for each of the methods (i.e., continuous simulation, flood frequency analysis, and event-based simulation) in Section 2.
(2)  Line 156 now explicitly states that "Flood frequency analysis is an important source of information when data are available and key assumptions (e.g. historical and future climatic and hydrological stationarity) are met, due to the implicit consideration of flood causing factors without a need for assumptions about joint interactions." acknowledging the key role in streamflow-based procedures in design flood estimation.
(3)  Lines 1040-1054 have been carefully edited to explicitly state the adequacy but also limitations of each method in the context of climate change. Line 1043 now states "Faulkner et al. (2020) advise the use of non-stationary flood frequency analysis as a means for obtaining current day estimates". We also acknowledge at Line 1044 that "In the case of continuous simulation, stochastically generating reliable rainfall sequences remains challenging (Woldemeskel et al., 2016)"

Pg.17-18. It is clear that a lot of effort went into producing the 'best estimates of central scaling rates' in which the authors independently assigned weights to different studies to arrive at a median estimate and 66% range. The Supplementary table is an impressive exercise in collating and trying to synthesize information from 179 different sources. However, it is a little difficult to see how the summary values in Table 1 were arrived at, since the weights are not provided and the thresholds shown in Table 1 (<1hr or >24hr) don't match those shown in Figure 2 (<6hr, >12hr) (perhaps I missed why Table 1 and Figure 2 differ). It would be helpful to provide more data on how the values were obtained, so it doesn't seem too subjective.

As per the reviewer's suggestion Table S3 now contains the quantitative results of each researcher and their methodology. The labelling on Figure 2 now matches Table 1 (1hr or 24hr). The grouping of the durations is now explained at Line 569: "Studies investigating storm durations of 6 hours or less were grouped into the hourly rainfall duration, with studies with durations of greater than 6 hours grouped with the daily rainfall duration."

**Minor comments**

L.99 "the primary differences between methods relating to where in the causal chain of flooding the data are obtained, and where the probability model is fitted" could be rephrased for greater clarity, perhaps being more specific.

This sentence was removed as design flood estimation is already defined in introduction.

L.139. The term "efficacy" in the caption of Figure 1 is a little ambiguous and could be clarified.

The word "efficacy" has been replaced with "utility".

L.172/183 "average effect size" – please specify the effect size of what.

This now states "magnitude of extreme rainfall change" (Line 193, 203).

L.177 "with variation to storm duration .. and location preserved" could be rephrased for clarity.

This now reads "Additionally, variation with storm duration, severity (i.e., AEP), and location were considered." (Line 197)

L.186 "was weighted by"?

This typo has been fixed.

L.206-211. Recent work has shown that groundwater is a more important driver of flooding than either antecedent soil moisture or antecedent extreme rainfall (see work by W.Berghuijs); this may be worth mentioning.

Line 225 now reads: "In the absence of snowmelt, changes in antecedent conditions related to soil moisture and baseflow have been shown to modulate flood events (Berghuijs and Slater, 2023)."

L.237-240 "even the use of physically-based covariates is problematic as the covariates should capture the differing processes": please clarify/elaborate. Also, which of the "statistical associations may not remain constant with climate change"?- it is worth being more explicit.

We now explain the different processes explicitly and state that the statistical relationships may change due to changes in the dominant flood mechanism. This text now reads (Lines 256-261): "But even the use of physically-based covariates is problematic as the covariates may not capture the differing processes that affect rainfall and therefore flood changes, for example thermodynamic versus dynamical changes to extreme rainfall which vary with storm duration (Schlef et al., 2018). A final complication is that even if the changes in flood drivers are captured by the covariates there is no guarantee that these flood drivers will be those governing flooding in the future due to changes in the dominant flood mechanism (Chegwidden, Oriana et al., 2020; Zhang et al., 2022; Wasko, 2022)."

L.432 "their application to the future period remains untested": the phrasing is a little odd; application to the future is always untested. Do you mean their predictive ability (to predict out of sample) is untested?

This is correct and the text has been changes to state "predictive ability" (Line 442).

L.957-961. I am not sure how helpful it is to tell the reader that the papers were assessed independently and through weighting of evidence, if the outcomes of those analyses are not presented. It's a bit like asking the reader to simply *trust* that the analysis is objective.

As per the reviewer's suggestion Table S3 now contains the quantitative results of each researcher and their methodology.

L.1068 "the impact on rare floods diminished" could be rephrased for clarity.

This has been rephrased to "a lesser impact on rarer floods" Line 1107.

**Figures**

Figure 1. Nice summary figure. The labels of the x-axis could explained in the caption. The "S" shape of the curve could also be explained.

Figure 2. Again, very interesting (and novel) summary figure. I don't think 2xCC and 3xCC are defined.

The caption of Figure 1 now explains the x-axis, notably that EY stands for "Events per year". The curve shape has been updated to be the more classical tapered curve. Finally, 1CC, 2CC, 3CC are all now labelled on Figure 2.